# Perceptual prediction error supports implicit process in motor learning

Xiaoyue Zhang[1], Wencheng Wu[iD][1], Kunlin Wei[iD][1,2]*

1 School of Psychological and Cognitive Sciences, Peking University, Beijing, China, 2 Beijing Key Laboratory of Behavior and Mental Health, Beijing, China

* wei.kunlin@pku.edu.cn

## Abstract

Error-based learning underlies motor learning, but what specific motor error drives implicit learning, the procedural component of motor skill, is unclear. A typical action consists of a movement and a performance outcome, e.g., grabbing a coffee cup involves a reaching movement and its actual landing of the hand relative to the target cup. While performance error is fundamental for the cognitive component of motor learning, what error, either performance or movement prediction error, underlies implicit motor learning has not been resolved. These two errors are hard to disentangle as the performance outcome is an integral part of the movement. Here we used the classical visuomotor adaptation paradigm, in which people learn to counter visual perturbations by deliberately aiming off the target, to dissociate the performance error from the prediction error. Using a series of behavioral experiments and model comparisons, we revealed that movement prediction error, but not performance error, can parsimoniously explain diverse learning effects. Importantly, despite the perturbation being visual, the movement prediction error is not specified in visual terms, but determined by a perceptual estimate of the hand kinematics. In other words, contrary to the widely-held concept of sensory prediction error, a perceptual prediction error drives implicit motor learning.

### Author summary

Everyday actions, like reaching for a cup of coffee, require precise motor control. When errors occur, the brain adapts through two distinct processes: a conscious strategy (explicit learning) and an unconscious, automatic adjustment (implicit learning). While it is widely accepted that explicit learning is driven by performance error, the specific error signal for implicit learning remains under debate. Prevalent theories attribute implicit motor learning to either performance error or sensory prediction error. Here, we propose a novel error, *perceptual prediction error* (PPE), derived from the brain's internal estimate of hand location, which integrates visual, predictive, and proprioceptive cues. Through a combination of

**Data availability statement:** Data and code are available at: https://osf.io/tjc52/.

**Funding:** This work was supported by the STI2030-Major Projects (2021ZD0202600 to KW) and the National Natural Science Foundation of China (32471099, 62061136001 to KW). The funders had no role in study design, data collection and analysis, decision to publish, or preparation of the manuscript.

**Competing interests:** The authors have declared that no competing interests exist.

behavioral experiments and computational modeling, we provide compelling evidence that PPE, rather than traditional visual errors, best explains the patterns of implicit learning under various manipulations, suggesting that PPE serves as the primary driver of implicit learning.

## Introduction

Learning from errors is a fundamental aspect of human cognition, particularly in the realms of perception and action [1–3]. A classic example is the basketball player's effort to improve their jump shot; when a shot lands left of the basket, the player consciously adjusts their aim to the right in subsequent attempts. This explicit learning is complemented by an implicit learning process, which involves subconscious fine-tuning of bodily movements without conscious correction or even awareness of these fine-tunes. While performance error is recognized as a key driver of explicit learning across various motor tasks [4–6], the specific errors that facilitate implicit learning—a defining feature of motor skill acquisition—are still a matter of intense debate.

The debate on motor errors centers around two primary categories: performance error or movement prediction error [7]. Performance error is typically defined as the discrepancy between movement outcome and the intended goal [8–13], while movement prediction error refers to the discrepancy between the executed movement and its predicted sensory consequences [14–18]. The complexity of dissociating these two errors arises from their simultaneous occurrence during action execution. For instance, when reaching for a coffee cup, the desired trajectory of the hand and the desired final position must both be considered [19]. The trajectory prescribes a movement prediction error, and the final position prescribes a performance error, both theoretically leading to implicit learning for subsequent movements [2,14,18,20].

Prominent computational models of motor learning [8–13] and foundational textbooks (e.g., [21]) have primarily focused on performance error (PE) to explain implicit learning behaviors. In contrast, a growing body of behavioral and neurophysiological research highlights the role of sensory prediction error (SPE) —the difference between the actual movement trajectory and the intended movement trajectory—as a critical factor in driving implicit learning [14,15,18,22–26]. Resolving this theoretical inconsistency in the understanding of implicit motor learning is not only essential for cognitive science [7] but also promises to translate algorithmic insights from human procedural learning into applied fields such as robotic control and artificial intelligence [27].

In this study, we harness the classical visuomotor rotation (VMR) adaptation paradigm —an established framework for investigating error processing in motor learning—to elucidate the competing error signals that contribute to implicit learning [18,20]. During the VMR task, participants encounter a hand cursor that moves in a skewed direction relative to their actual hand motion, prompting them to re-aim their movements away from the intended target. This dissociation between movement prediction (aiming) and goal attainment (cursor motion) allows us to separate

PE and SPE within a single action (Fig 1A): PE is the deviation of the final cursor position from the target, while SPE is characterized by the deviation of the visual cursor motion from the re-aiming movement. Importantly, SPE is based solely on visual feedback, with the implicit assumption that the motor system adapts itself based on the visual consequence of actions, i.e., the skewed cursor motion. However, spatially locating one's moving body effector is the foundation of motor control [30–32]. We thus propose a novel third category of error: perceptual prediction error (PPE), which represents the discrepancy between the *perceived* hand movement from the predicted re-aiming movement. Previously, the discrepancy between the perceived hand movement and the intended movement has been proposed to drive error-clamp adaptation [29,33], a special case of VMR where explicit learning is absent. However, in the error-clamp adaptation, the predicted direction and the target direction are spatially aligned, and thus PE and PPE are indissociable. In VMR adaptation with explicit learning, the re-aiming direction naturally dissociates these two errors, enabling an opportunity to critically examine their roles.

Here we designed a series of six experiments that systematically manipulated VMR perturbations, revealing distinct characteristics of implicit motor learning. Through comprehensive behavioral analyses and modeling, we present compelling evidence that perceptual prediction error, rather than performance error or sensory prediction error, serves as the primary driver of implicit motor learning. This finding offers a parsimonious explanation for our new results while reconciling them with established literature in the field.

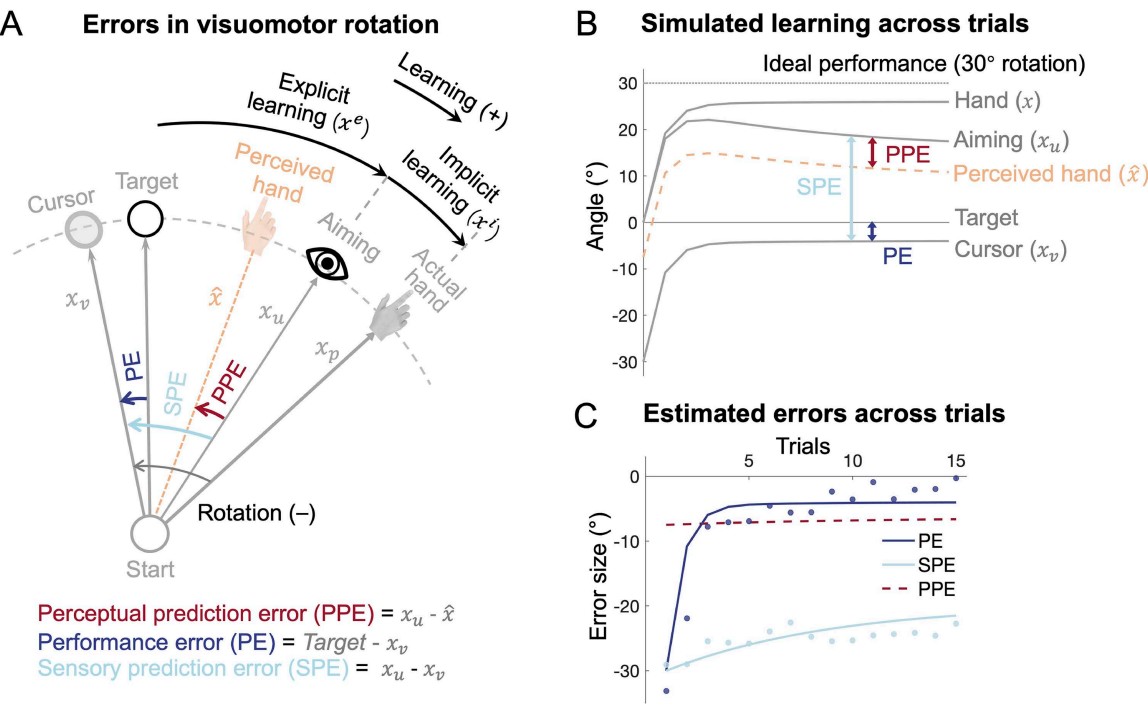

**Fig 1. Illustration of sensorimotor cues and error dynamics in visuomotor rotation (VMR) learning. (A)** Schematic representation of the key sensory cues and errors during VMR learning. The rotated visual cursor, actual hand position, and re-aiming direction correspond to the visual cue ($\mathbf{x}_v$), proprioceptive cue ($\mathbf{x}_p$), and sensory prediction ($\mathbf{x}_u$), respectively. Performance error (PE) is the discrepancy between the cursor and the target, sensory prediction error (SPE) is the discrepancy between the cursor and the sensory prediction, and perceptual prediction error (PPE) is the discrepancy between the prediction and the perceived hand direction ($\hat{\mathbf{x}}_{hand}$) combination, computed from a Bayesian integration of $\mathbf{x}_v$, $\mathbf{x}_p$, and $\mathbf{x}_u$. **(B)** Simulation of a 30° VMR learning using the PPE model. The time series of the variables from (A) are shown, with solid lines representing variables that are directly observable, and the dashed line represents the perceived hand direction ($\hat{\mathbf{x}}_{hand}$), which is inferred but not directly observable. **(C)** Error dynamics during learning under a 30° VMR perturbation. Simulated PE and SPE match well with the observed data [28] represented by the dots. Distinctly, simulated PPE remains small and stable, but quantitatively matches our previous estimation of this error [29].

## Results

### Errors and models

In the VMR paradigm, participants need to move a directionally rotated cursor toward the target, and the directional deviation of the cursor ($x_v$) from the target defines the PE (Fig 1A). PE drives the participants to adapt their movements in the subsequent trial by consciously re-aiming away from the target in the opposite direction of the rotation perturbation [34]. This re-aiming reflects the participant's prediction of where their hand will move, generating a sensory prediction of the reaching action ($x_u$). The deviation between the actual cursor motion and this prediction is regarded as a sensory prediction error, given its purely visual nature [7,18]. However, we propose that since the predicted variable pertains to the intended *hand* direction, the relevant feedback should also concern the hand rather than the cursor [29]. Thus, the perceived hand ($\hat{x}_{hand}$), rather than the cursor, is compared to the predicted hand ($x_u$), with their discrepancy constitutes a perceptual prediction error. The perceived hand ($\hat{x}_{hand}$) is derived from a combination of multiple sensory cues, including the visual cue ($x_v$) from the cursor, the proprioceptive cue ($x_p$) from the actual hand, and the motor prediction cue ($x_u$). These cues are integrated in accordance with Bayesian principles, with each cue weighted by its relative reliability. Notably, motor prediction can not only provide a reference for error computation, but also serve as an internal predictive cue that is integrated with sensory feedback cues [35–38].

The three types of errors—PE, SPE, and PPE—exhibit distinct temporal profiles throughout the learning process, such as during a 30° counterclockwise VMR (Fig 1B and 1C). When the perturbation is introduced, the cursor initially deviates 30° from the target, and through trial-by-trial gradual learning, the cursor homes back toward the target. This results in an exponential decrease in PE, driven by the gradual correction of hand movement direction. In contrast, re-aiming behavior—which reflects explicit learning—initially rises sharply and then exhibits a small decline over the course of learning [28]. The re-aiming behavior dictates the dynamics of the prediction errors (Fig 1C): SPE remains large throughout the learning process because the re-aiming direction stays consistently offset from the cursor. In contrast, PPE remains relatively small and stable during learning, as the re-aiming direction is close to the perceived hand direction, which is derived from a combination of sensory cues ($x_u$, $x_v$, and $x_p$; Fig 1A).

To determine which error best explains the diverse learning phenomena observed in motor adaptation, we implemented a unified model framework. State-space models, which have been successfully used to model error-based learning in visuomotor adaptation [8,10,39,40], were applied to test whether implicit learning is driven by PE, SPE, or PPE (see Methods for model details).

The perceptual estimation of hand location is contingent on the relative reliability of the individual sensory cues. Importantly, visual uncertainty increases with larger SPEs [29], as participants tend to fixate not on the target but closer to their aiming direction during the movement (S1 Fig; [41], [42]), as shown in our eye-tracking experiment (S1 Fig). This results in the cursor deviating from the fixation point by an angular separation equivalent to the magnitude of the SPE. Our previous work demonstrated that visual uncertainty regarding cursor motion increases linearly with this eccentricity [29]. We also confirmed that a nonlinear, instead of a linear, eccentricity effect on visual uncertainty would not affect model comparison results (S2 Fig and S7 Table). While a large visual perturbation induces both large SPE and PE (at least during early learning; Fig 1C), its effect on PPE is dampened due to the high visual uncertainty associated with large perturbations. Visual uncertainty reduces the influence of the visual feedback on the perceived hand direction ($\hat{x}_{hand}$) according to Bayesian principles of perception [43]. By incorporating the perceptual estimate of hand direction into the prediction error, the dynamics of PPE diverge significantly from those of PE and SPE, thus predicting implicit learning differently, as we show below in a series of experiments.

**Experiment 1: Perturbation size-dependency of implicit learning reflects the dynamics of PPE, not PE or SPE.** Manipulating the size of visual perturbations is a straightforward way to probe error-based learning and to differentiate between distinct error signals. In VMR, larger perturbations increase overall and explicit learning, but

implicit learning remains stable across a broad range of perturbation sizes [16,28,44,45]. This classical finding has been interpreted as a hard limitation of implicit learning [20,33,28,44] or a result of competition of explicit and implicit learning, where the explicit learning "siphons" away performance error [10].

The three error types—PE, SPE, and PPE—behave differently during early learning. Both PE and SPE increase with larger perturbations, but PPE remains small (Fig 2B). Previous studies have often overlooked early learning, focusing on late learning [10]. We examined the entire learning process using various perturbation sizes (15°, 30°, 60°, and 90° VMR) by analyzing a prior dataset first [28] (Fig 2A). To show the distinct size-dependency and dynamics throughout learning, we calculated the PE and SPE from empirical data and estimated PPE from PPE model predictions (Fig 2B). As expected, larger perturbations led to an increase in both PE and SPE during early learning, but the scaling effect maintains during late learning only for SPE, not for PE (Fig 2B, middle and right; Fig 2D). Remarkably, implicit adaptation did not scale with perturbation size. Instead, it remained unchanged or even slightly decreased, both in early and late learning (Fig 2E; one-way ANOVA, early: $F(3,36) = 0.2$, $p = 0.897$; late: $F(3,36) = 1.44$, $p = 0.247$). Neither the PE nor SPE model captured this pattern, as both models predicted a linear increase in early implicit learning with perturbation size. The SPE model also incorrectly predicted a linear increase in late implicit learning (Fig 2E). In contrast, the PPE model successfully explained across perturbation sizes for both early and late stages (Fig 2C and 2E), as PPE remains largely invariant across perturbation sizes (Fig 2B left and 2D). Model comparison showed that the PPE model significantly outperformed both the PE and SPE models ($R^2 = 0.708$, 0.177, and -0.730, BIC = 182.04, 260.69, and 528.63 for PPE, PE, and SPE models, respectively; S1 Table). Note here we fitted all three models to the observed implicit learning data with identical procedures, ensuring that the models are evaluated on equal footing.

The fact that PPE does not exhibit a linear increase with perturbation size, unlike PE and SPE, stems from the nature of perceptual estimation. PPE is based on the perceived hand direction, which integrates visual, proprioceptive and motor prediction cues, each weighted by its uncertainty. Prior models based on PE and SPE have overlooked the fact that visual uncertainty increases with perturbation size [29]. During VMR learning, participants tend to fixate near their re-aiming direction, not on the target [41,42]. We confirmed this through a control experiment where we measured eye fixations during VMR learning (S1 Fig). Larger perturbations had larger explicit learning with fixation further away from the target. This greater angular separation between the cursor and the fixation point increases visual uncertainty, reducing the influence of visual perturbation on hand localization ($\hat{x}_{hand}$) according to Bayesian principles [29]. By incorporating visual uncertainty into the PPE model, we captured the classical finding of invariant implicit adaptation over perturbation size. This result suggests the apparent limit on implicit learning is not a reflection of a fundamental constraint in the motor learning system, nor is it due to competition for performance error, but a result of changing PPE.

The trial-by-trial aiming reports in *Bond & Taylor., 2015* enabled quantification of PE and SPE throughout learning. However, frequent aiming reports can alter the composition of explicit and implicit learning ([34, 28, 46, 47]). To minimize this potential confound, we conducted Experiment 1, which largely replicated Bond et al.'s study but removed the trial-by-trial aiming reports. Instead, we inserted a small number of exclusion trials during late learning to specifically measure implicit learning [47–49]. Additionally, the PPE model predicts a biased perceived hand localization after each movement, which cannot be directly measured. In Experiment 1, we measured this bias indirectly by requiring the participants to judge the location of their passively moved hand immediately after an adaptation trial [29] (Fig 3A). People's judgment should be biased in the direction of the visual perturbation, an effect termed as proprioceptive bias [49–54]. Importantly, we hypothesized that the magnitude of proprioceptive biases should correlate with the PPE and the implicit learning estimated by the model.

Experiment 1 recruited four independent groups of participants (n = 15 per group) to adapt to 15°, 30°, 60°, or 90° VMR (Fig 3A). The overall learning patterns were consistent with previous studies [28,45,55]. Implicit learning, measured through exclusion trials during late adaptation, exhibited a slightly nonlinear dependency on perturbation size (Fig 3D). A one-way ANOVA revealed a significant main effect of perturbation size in implicit learning ($F(3,56) = 7.15$, $p = 0.0004$).

**Fig 2. Behavioral and model fitting results from _Bond & Taylor, 2015_.** (A) Total, explicit, and implicit learning during VMR adaptation with varying perturbation sizes. Explicit learning was estimated from participants' reported aiming directions. Solid lines represent the mean, and shaded areas depict the standard error (SEM) for each perturbation condition. (B) Changes in perceptual prediction error (PPE), performance error (PE), and sensory

prediction error (SPE) during learning. PE and SPE were computed from data, while PPE was estimated through model fitting. The three errors show distinct temporal profiles, particularly during early learning. **(C)** Model fits to implicit learning across perturbation sizes, with explicit learning treated as known data. The PPE model accurately replicated the implicit learning patterns. In contrast, the PE model performed poorly for large perturbations (60° and 90°), and the SPE model struggled with small perturbations (15° and 30°). **(D)** The distinct patterns of PPE, PE, and SPE as a function of perturbation size during early (solid lines) and late (dashed lines) learning. While PE and SPE scale with perturbation sizes, PPE remains relatively stable across perturbations. **(E)** Data and model predictions for early (left) and late (right) implicit learning. Both early and late learning show minimal scaling across perturbation sizes, in contrast to the predictions of the PE and SPE models. The model predictions for implicit learning reflect the error dynamics illustrated in **(D)**. Error bars represent SEMs for the data and bootstrap-derived standard deviations for the model.

Post-hoc tests showed that implicit learning was significantly smaller for the 90° perturbation compared to both 15° ($p = 0.046$) and 30° ($p < .0001$), suggesting a concave size-dependency when continuous aiming reports were not required.

Despite reasonable fits to total and explicit learning (S3 Fig), the PE and SPE models incorrectly predicted a scaling effect of implicit learning with perturbation size. This discrepancy arose PE and SPE scaled with the perturbation size (Fig 3B), as observed in Bond & Taylor's study (Fig 2B). In contrast, the PPE model accurately predicted implicit learning, showing that PPE first increased and then decreased with perturbation size, leading to a concave pattern that matches the empirical data (Fig 3B and 3C, $R^2 = 0.969$; S2 Table). The model's parameters aligned with previous estimates, with implicit and explicit learning rates ($B_i = 0.151$ and $B_e = 0.513$) comparable to previous studies [8,9]. The slope of visual uncertainty with perturbation size ($k = 0.204$) was slightly lower than that in the error-clamp paradigm ($k = 0.31$; [29]), likely because participants in the VMR paradigm focus more on the cursor to ensure accurate performance. The results of our eye-tracking experiment supported that participants tended to fixate back on the cursor after completing their movements (S1B Fig). This contrasts with the error-clamp paradigm, where participants are instructed to ignore the cursor and fixate on the target, resulting in higher visual uncertainty. We also conducted two complementary analyses, i.e., parameter recover and confusion matrix analysis for model discriminability, to further demonstrate the reliability and robustness of our modeling results (see S4 and S5 Figs).

The proprioceptive biases significantly correlated with implicit learning, as predicted by the PPE model. During late adaptation, participants' right hands, the hand used for reaching adaptation, were passively moved to a random position near the target, and they were asked to indicate the direction of the right hand using their left hand. This passive localization test revealed a proprioceptive bias towards the visual perturbation, a phenomenon known as proprioceptive recalibration [29,49,50,52,56]. Importantly, the bias followed a concave trend across perturbation size, mirroring the pattern of implicit learning (Fig 3D). A one-way ANOVA revealed a significant main effect of perturbation size ($F(3,56) = 3.700$, $p = 0.017$), with a post-hoc test revealing a higher bias for 30° than for 90° perturbations ($p = 0.011$). A correlation analysis across participants showed a significant positive correlation between proprioceptive bias and implicit adaptation (Fig 3E; $R = 0.41$, $p = .001$). This high correlation aligns with the PPE model, as both measures are driven by the misperceived hand direction under visuomotor rotation ($\hat{x}_{hand}$, Fig 1). In the proprioception test trial, the proprioceptive judgment is biased by the misperception of the hand location experienced after the preceding adaptation trial ends [29], resulting in a small but consistent perceptual bias in the direction of visual perturbation.

In summary, implicit learning exhibited either a concave size-dependency (Experiment 1; [55]) or invariance [28] across perturbation sizes, depending on the task specifics, such as the presence of frequent aiming reports. Neither the PE nor SPE model could account for these size-dependencies, as their error sources did not follow a similar size-dependency (Figs 2D and 3B). Interestingly, although PE converges a small value across different perturbation sizes during late learning, enabling the PE model to capture late learning size dependencies [10], it fails to account for early learning, where it erroneously predicts a size-scaling effect. Only the PPE model provides a parsimonious explanation of the size dependencies of implicit learning across both early and late learning phases (Figs 2F and 3C).

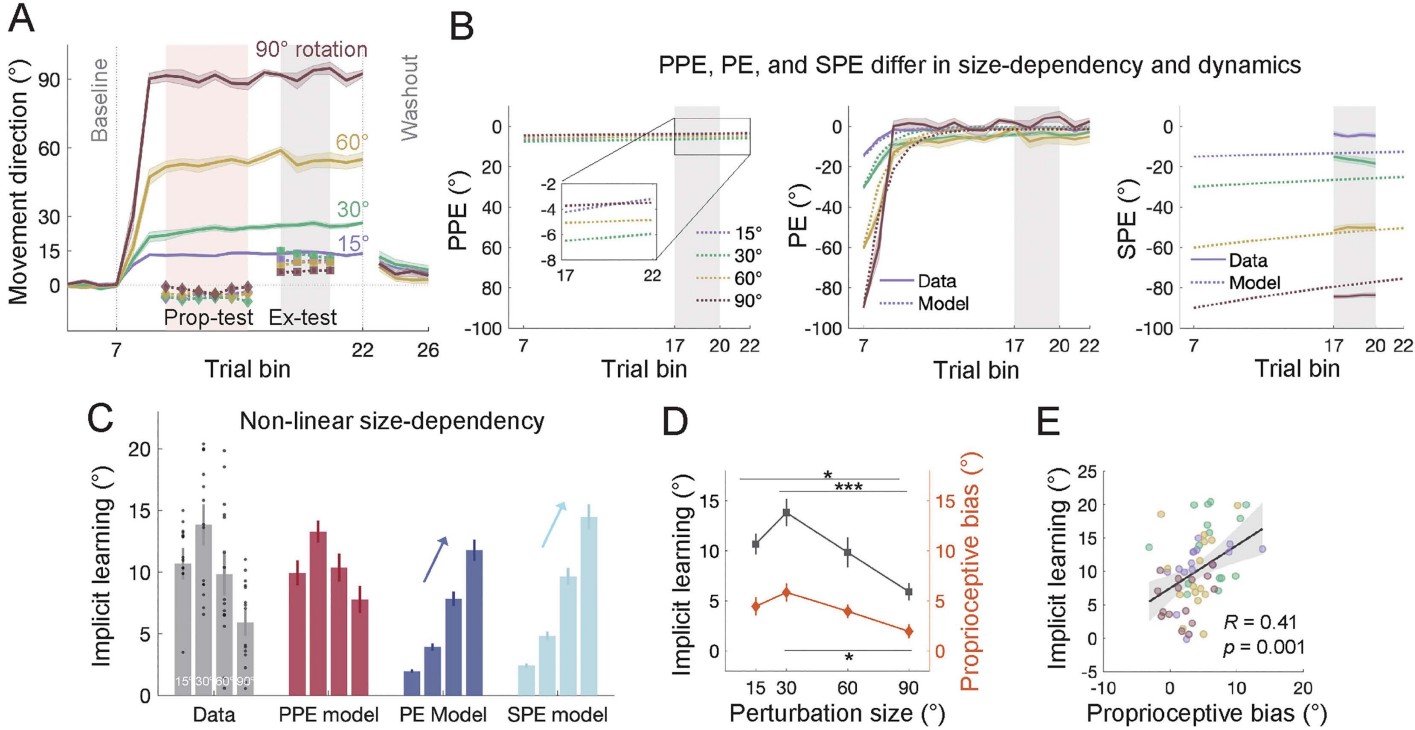

**Fig 3. Results of Experiment 1. (A)** Learning curves for VMR across different perturbation sizes. Learning starts after baseline, gradually reaches a steady-state, and then decreases significantly during the no-feedback washout phase. Colored lines represent the mean, and shaded areas represent the SEM for each condition. Light pink and gray shading regions indicate the timing of proprioception test trials ("Prop-test") and exclusion trials ("Ex-test"). The group average and SEM for implicit learning and proprioceptive bias are shown. Proprioceptive judgements are biased in the direction of the visual perturbation, which is opposite to the direction of learning. **(B)** Time course of the three error types, with data shown as solid lines and model estimates as dashed lines. Shading denotes SEM. Similar to *Bond & Taylor, 2015*, perturbation size scales the early PE and overall SPE, but not PPE. **(C)** Model fitting results for implicit learning. Only the PPE model accurately predicted the concave relationship observed in the data, whereas both PE and SPE models erroneously predicted a linear scaling effect. Error bars represent SEM for the data and bootstrapped standard deviations for the model. The black dots represent individual data. **(D)** Implicit learning and proprioceptive bias as a function of perturbation size. Both exhibited a similar nonlinear relationship (* denotes $p < .05$ and *** denotes $p < .001$). Note that the *proprioceptive bias* has been sign-flipped here to facilitate comparison with implicit learning. **(E)** Implicit learning was significantly correlated with proprioceptive bias on the individual level, consistent with the predictions of the PPE model. Each dot denotes the two variables measured in each condition.

**Experiment 2: Stepwise perturbations enhance implicit learning due to increased PPE.** The differences between the three competing errors are most pronounced at the time when a perturbation is initially introduced (Figs 1C, 2B and 3B). Therefore, if the perturbation size changes *during* learning, we expect these errors to have diverging effects. A previous study has shown that, compared to a single large perturbation (e.g., a 60° VMR, referred to as one-step perturbation herein), incrementally increasing perturbations (e.g., a 15→30→45→60°, referred to as stepwise perturbation) boosts implicit learning while reducing explicit learning [57]. The reduction in explicit adaptation is expected due to smaller PE in stepwise conditions. However, the increase of implicit learning challenges the conventional view that implicit learning is stereotypical and unaffected by various manipulations [16,28,44,45], and its mechanism remains unclear. Importantly, stepwise adaptation offers a critical test of the three error models. For both PE and SPE, one-step perturbation generates larger errors than stepwise perturbation during early learning, predicting greater implicit learning for the one-step condition (Fig 4B, middle and right). In contrast, the PPE model predicts the opposite: smaller PPE for one-step perturbation due to the discounting effect of visual uncertainty, and thus, less implicit learning (Fig 4B, left).

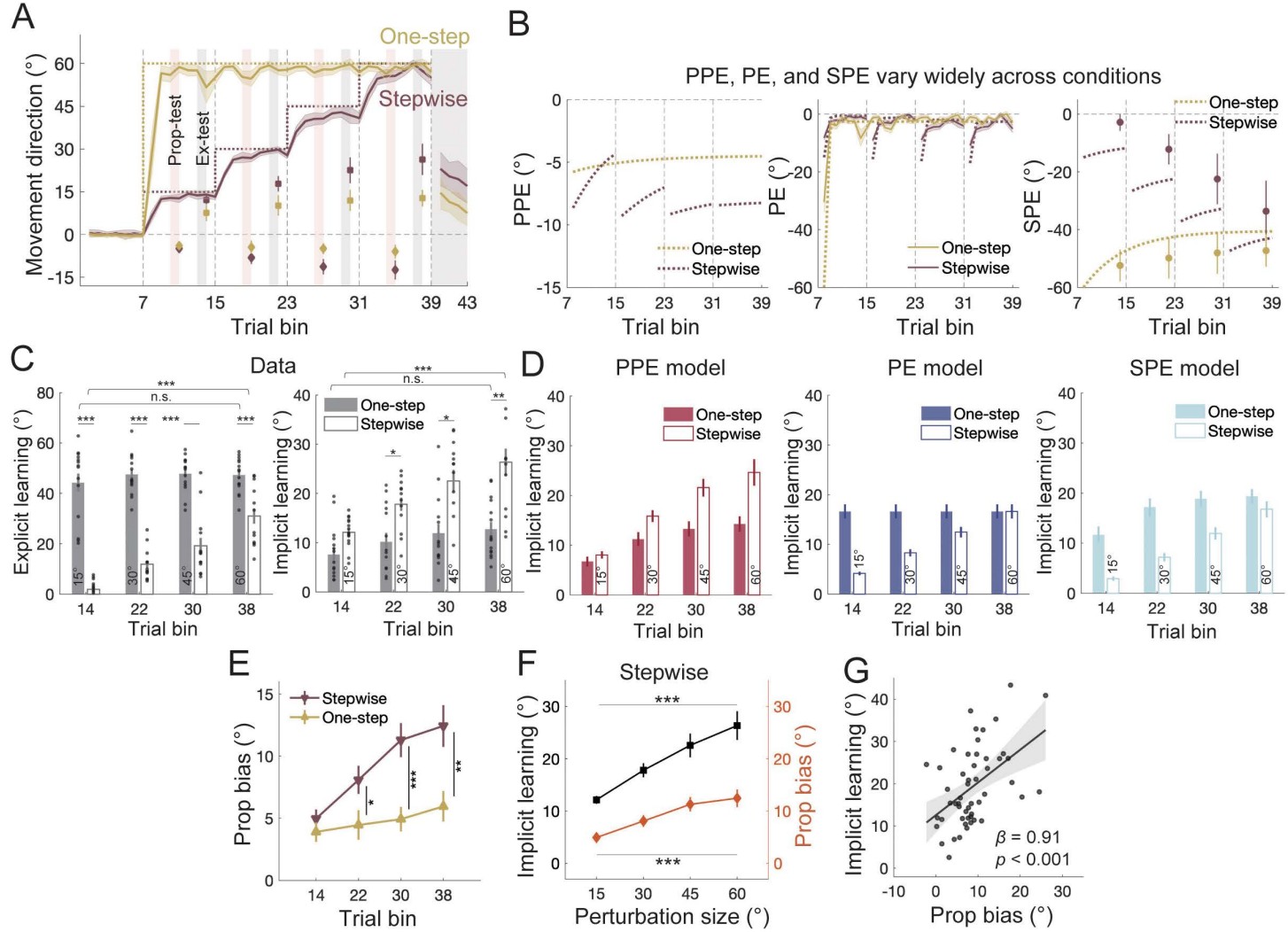

**Fig 4. Results of Experiment 2. (A)** Learning of the one-step (yellow) and stepwise (brown) group. The light pink shading indicates the timing range for the proprioception test trials, and the light grey shading indicates the timing range for the exclusion trials. Squares with error bars represent implicit learning measured via exclusion trials ("Ex-test"), while diamonds with error bars represent proprioceptive bias measured by proprioception test trials ("Prop-test"). **(B)** Time courses of the errors. The solid lines and dots represent data, and the dashed lines represent model estimates. PE and SPE were larger for one-step than for stepwise perturbations during early learning, while PPE showed the opposite condition difference. **(C)** Time courses of explicit and implicit learning. Both explicit and implicit learning increased over time for stepwise adaptation, but remained unchanged for one-step adaptation. Note the difference between conditions: while explicit learning was larger for one-step adaptation, implicit learning was larger for stepwise adaptation especially during late learning (* denote $p < 0.05$, ** denote $p < 0.01$, and *** denote $p < 0.001$). Error bars are SEMs for the data. The black dots represent individual data. **(D)** Model fittings for implicit learning. Error bars are bootstrap-derived standard deviations for the model (resample size = 5000). Only the PPE model replicated the actual learning patterns, while both PE and SPE models predicted one-step adaptation was larger than stepwise adaptation in most cases. **(E)** Time course of proprioceptive bias in the two conditions. The sign of proprioceptive bias was flipped to aid the comparison. With larger PPE, stepwise adaptation elicited a larger proprioceptive bias than one-step adaptation. **(F)** Implicit learning and proprioceptive bias plotted as a function of perturbation size during stepwise adaptation. **(G)** Implicit learning was significantly correlated with proprioceptive bias on the individual level. Each dot denotes the two variables measured in each perturbation size during stepwise adaptation.

To test these predictions in Experiment 2, we recruited two groups of participants: one group experienced a one-step 60° VMR perturbation (n = 15), while the other group experienced a stepwise 15°→30°→45°→60° VMR perturbation (n = 15; Fig 4A). Implicit learning was assessed using exclusion trials during late learning. Both groups achieved similar

steady-state performance by the end of learning (two-sample $t$-test, $t(28) = -1.117$, $p = 0.274$), but their implicit learning patterns diverged (Fig 4C). The stepwise group showed increasing implicit learning with each perturbation step ($F(3,21) = 11.639$, $p < 0.001$), while the one-step group showed no significant change in implicit adaptation over time ($F(3,21) = 1.938$, $p = 0.154$). Notably, from the second exclusion trial block onward, the stepwise group exhibited significantly greater implicit learning than the one-step group ($p = 0.011$). By the final exclusion trial block (at 60° VMR), implicit learning in the stepwise group was nearly double that of the one-step group ($26.33° \pm 2.74°$ vs. $12.74° \pm 1.45°$; $p = .001$). In contrast, explicit learning was consistently larger for the one-step group for each exclusion trial block ($p < 0.001$).

All three models predict the gradual increase in implicit learning with stepwise perturbations, as all errors increase incrementally (Fig 4B and 4D). However, only the PPE model correctly captures the relatively smaller implicit learning for the one-step perturbation compared to stepwise adaptation (Fig 4D). The PE and SPE models erroneously predict the opposite. Model comparison confirmed the superior performance of the PPE model ($R^2 = 0.960$, $0.933$, and $0.925$; BIC $= 264.81$, $309.59$, and $317.67$ for PPE, PE, and SPE models, respectively; more model comparisons, see Table A in S3 Table). The superior performance of the PPE model was obtained even though we used the visual uncertainty parameter ($k$) estimated from Experiment 1, which also involved a one-step perturbation.

As in Experiment 1, proprioceptive bias in this experiment correlated with implicit learning, further supporting the PPE model. Proprioceptive bias increased with stepwise perturbation (Fig 4E, $F(3,25) = 8.874$, $p < 0.001$), but remained unchanged during the one-step perturbation ($F(3,25) = 0.538$, $p = 0.661$). These proprioceptive bias patterns mirrored those of PPE and implicit learning (Fig 4F). To analyze the individual difference, we used a mixed linear model to isolate the between-subject variability and revealed a significant positive correlation between proprioceptive bias and implicit learning ($\beta = 0.91$, $p < 0.001$; Fig 4G). This result was consistent with previous studies on proprioception and motor adaptation [29,33,50,52], supporting that implicit adaptation and proprioceptive bias are driven by the same error source, i.e., PPE.

One might argue that the one-step perturbation amplifies explicit learning compared to the stepwise perturbation, while the model-based analyses above assume fixed explicit learning parameters across both conditions. Indeed, when faced with a large, salient perturbation, individuals may rely more heavily on explicit learning strategies [10,57]. To address this, we allowed the explicit learning parameters to vary between the two conditions and re-evaluated the models (S6 Fig and Table B in S3 Table). As expected, the explicit learning rate was higher in the one-step condition across all models. However, even with varying explicit learning parameters, both the PE and SPE models still failed to account for the observed differences between conditions (S6 Fig) and yielded unrealistic model parameters (Table B in S3 Table; [8,9,58]). In contrast, the PPE model continued to provide excellent fits across both conditions, yielding the best overall fit (S6 Fig and Table B in S3 Table).

We further validated the models using cross-validation, fitting each model with data from the stepwise group and testing it on the one-step group. Again, the PPE model outperformed the other two models with a large margin in cross-validation (S6C Fig and Table C in S3 Table).

**Experiment 3: Suppression of explicit learning uncovers nonlinear implicit learning compatible only with PPE.** When a visual perturbation is first introduced, PE and SPE equal the perturbation size, as the participant has not yet developed a re-aiming strategy. In contrast, PPE is less than the visual perturbation because it derives a multisensory cue combination for hand localization (Fig 1A). Thus, the learning to the very first perturbation can highlight the key differences between errors: both PE and SPE predict a linear learning function relative to perturbation size, while PPE predicts a nonlinear function, as larger perturbations increase visual uncertainty, reducing its own influence on hand localization [29,59].

In Experiment 3 (n = 18), we used trial-by-trial random perturbations to suppress explicit learning and isolate the single-trial implicit learning. Unlike blocked perturbations as in Experiments 1 and 2, random perturbations prevent the accumulation of implicit learning. The participants performed single-target reaches, but each perturbation trial was

perturbed by a visuomotor rotation with a pseudo-random size and direction (0, ±4°, ±8°, ±16°, ±32°, and ±64°), and was flanked by two no-feedback null trials ([Fig 5A]). Participants were instructed to aim directly at the target in the null trials, thus discouraging explicit learning. Implicit learning was measured by comparing hand movement directions between the pre- and post-perturbation null trials [29]. The perturbation sequence, though it appeared random, was identical for all participants. The perturbation sequence had a zero mean, allowing any residual explicit learning to average out.

The participants displayed adaptive changes immediately after the perturbation sequence was introduced, as evidenced by increased variability in hand directions ([Fig 5B]). We fit the learning data with the three competing models. Since PE and SPE are equal to the perturbation size in single-trial learning, we compared only the PPE model and PE model. The PPE model outperformed the PE model in explaining the learning response to random perturbations ([Fig 5B], $R^2 = 0.471$ vs -0.683; BIC = 46.56 vs 82.00). As predicted by the PPE model, implicit learning follows a nonlinear pattern: it increased with perturbation size until a point, then perturbation size grew larger ([Fig 5C]; repeated-measure ANOVA: $F(3.2,53.8) = 45.897$, $p < .0001$; 4° vs 8°: $p = 0.005$; 8° vs 32°: $p = 0.001$; 32° vs 64°: $p < .0001$). This nonlinear pattern was consistent with previous findings from various motor adaptation paradigms [29,33,55,59]. In contrast, the PE (SPE) model predicted a linear pattern (linear regression: $y = a + b\theta$, with $a = -0.007$ and $b = -0.064$, $R^2 = 0.908$, $F = 89.1$, $p < .0001$), that drastically deviated from the observed data ([Fig 5C]; $R^2 < 0$, $RMSE = 2.449°$), while the PPE model captured the nonlinear

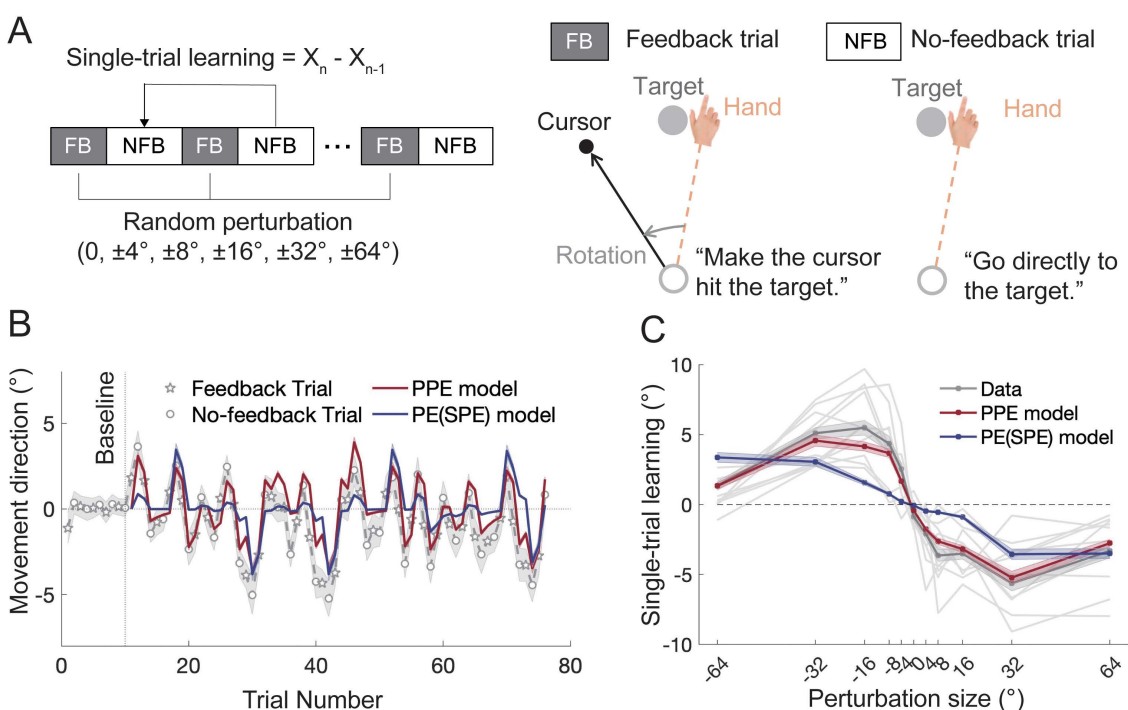

**Fig 5. Results of Experiment 3. (A)** Experimental design. In feedback (FB) trials, participants experienced visuomotor rotations with pseudo-random perturbation sizes and directions (0, ±4°, ±8°, ±16°, ±32°, and ±64°). Each feedback trial was flanked by two no-feedback (NFB) null trials, where participants were instructed to move directly to the target. Single-trial implicit learning was measured by the directional change between pre- and post-perturbation null trials. **(B)** Time course of hand movement directions across trials, along with model fits. The grey line (with shaded SEM) represents the data, with stars denoting feedback trials and circles representing no-feedback trials. The PPE model (red) provided a better fit to the data than the PE/SPE model (dark blue). **(C)** Single-trial learning as a function of perturbation size, derived from both trial-by-trial data and model fits. The PE/SPE model predicted a linear relationship between perturbation size and learning, while the PPE model predicted a nonlinear relationship. Color shades represent SEMs for the data and bootstrapped standard deviations for the model. The light grey lines represent individual participants' data.

pattern well ($R^2 = 0.954$, $RMSE = 0.684°$). Despite the randomness of the perturbations, the slope of visual uncertainty ($k$) estimated in this experiment was consistent with that from the block-design experiments (S7 Fig), further supporting the robustness of the PPE model across different experimental designs.

Despite the task instructions, participants might have strategically re-aimed away from the visual target in the null trials, potentially "contaminating" the estimates of implicit learning with explicit learning. To assess this, we employed a model-based analysis to include an explicit learning component. The results from the PPE model revealed that the estimated explicit learning parameters were quite small (learning rate $B_e = 0.047$ and retention $A_e = 0.037$), indicating a negligible contribution from explicit learning. Additionally, incorporating explicit learning into the model only marginally increases the $R^2$ but results in a higher BIC for the PPE model ($R^2 = 0.471$ vs. 0.500; BIC = 46.56 vs. 52.82). When we constrained the lower limit of $A_e$ to typical values (> 0.5), the inclusion of explicit learning actually worsened the model fit ($R^2 = 0.471$ vs. 0.453, BIC = 46.56 vs. 57.26). Similarly, the fit of the PE/SPE models deteriorated when explicit learning was added (S4 Table). Notably, the PPE model still substantially outperformed the PE and SPE models even with the inclusion of explicit learning (S4 Table). Overall, we conclude that random perturbations primarily lead to implicit learning with a nonlinear size dependency, which can only be explained by the PPE model.

**Experiment 4: Implicit learning elicited by instructed re-aiming is driven by PPE.** One of the seminal studies in motor neuroscience supporting the role of SPE employed the instructed re-aiming paradigm in VMR [15]. In this paradigm, participants are instructed to re-aim at a neighboring target whose angular deviation from the original target matches the imposed visual perturbation (e.g., 45°; Fig 6A). This re-aiming nullifies performance error (PE) immediately. Intriguingly, despite this, participants begin to drift away from the re-aiming target involuntarily over repeated trials, increasing PE. This drift has been interpreted as evidence that SPE drives implicit learning [15], as PE-based theories cannot explain the implicit drift when PE is nullified. Recently, Albert et al. (2022) proposed an alternative explanation based on the PE framework by introducing a new PE formulation (PE$_{re-aiming}$)—a form of PE that arises between the cursor and the re-aiming target rather than the original target (Fig 6B). Notably, this PE$_{re-aiming}$ is equivalent in magnitude to SPE, thus enabling PE-based models to account for the drift phenomenon. Our PPE model can explain this drift without requiring any new concepts: the perceived hand position is "pulled" toward the visual cursor, deviating from the re-aiming target, thus generating a PPE that drives implicit learning. Model fits to the previous dataset [15] indicate that all three error models (PE, SPE, and PPE) can account for the original findings (Fig 6C). However, the models make distinct predictions when the re-aiming offset size changes. If the angular separation between the re-aiming and original targets increases (e.g., from 45° to 90°), both SPE and PE$_{re-aiming}$ would predict a larger implicit drift, as the errors increase with the offset. In contrast, the PPE model predicts a reduction in implicit drift because larger visual perturbations reduce the influence of PPE due to increased visual uncertainty.

To test these diverging predictions, Experiment 4 recruited two groups of participants who adapted to either a large (90°, n = 15) or small (45°, n = 15) re-aiming offset. Upon re-aiming, participants' movements began to drift in the opposite direction of the perturbation. Critically, the drift for the 90° group was significantly smaller than that for the 45° group (Fig 6E). The final drift in the 90° condition was considerably smaller than in the 45° condition (13.17° ± 1.29° for 45°, 8.44° ± 1.22° for 90°; two-sample $t$-test: $t(28) = 2.58$, $p = 0.016$). The PPE model readily explains this condition difference, as it predicts smaller implicit drift for larger re-aiming offsets due to the reduced influence of visual perturbations on hand localization. In contrast, both the PE and SPE models incorrectly predict that the drift should increase in the 90° condition (Fig 6E and S5 Table). The visual uncertainty parameter ($k$) estimated from this experiment (0.211) was similar to those in previous experiments, further supporting the robustness of the PPE model. In addition, we conducted parameter recovery and confusion-matrix analyses, as we did in Experiment 1. The results showed high concordance between simulated and recovered parameters (CCC > 0.7 for all parameters, S4 Fig), and strong discriminability with perfect model recovery achieved across all three alternative models (S5 Fig), demonstrating the reliability and robustness of our modeling approach.

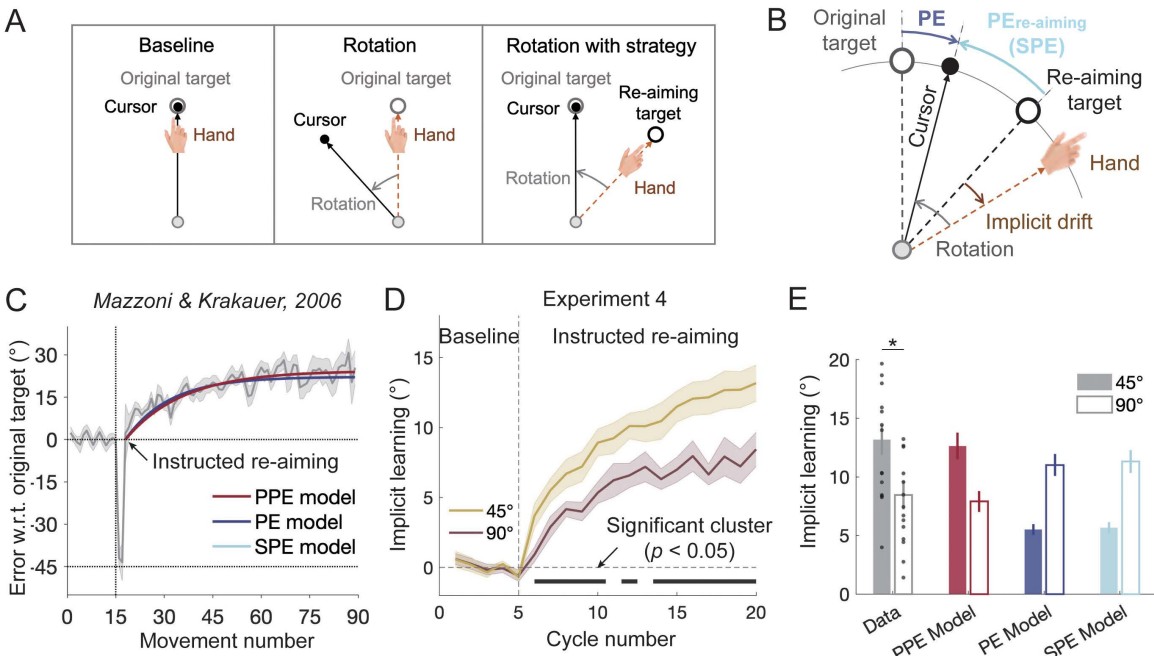

**Fig 6. Data and model fitting for the instructed re-aiming paradigm [15] and Experiment 4. (A)** Instructed re-aiming paradigm illustration. Participants initially moved a cursor toward a target without perturbation during baseline trials, followed by a single rotation trial with a 45° visuomotor rotation. During subsequent instructed re-aiming trials, they were instructed to move their hand toward a neighboring "re-aiming target" 45° away, allowing the cursor to perfectly hit the original target. **(B)** Motor errors during instructed re-aiming. Despite re-aiming at the new target, participants' hands unknowingly "drift" away from the re-aiming target. In addition to the three errors shown in Fig 1A, a new error termed re-aiming performance error ($PE_{re-aiming}$)—the deviation of the visual cursor from the re-aiming target—is introduced [10,26]. This error is equivalent to SPE. **(C)** Performance relative to the original target before and after instructed re-aiming. Data from Mazzoni & Krakauer (2006). After the initial 45° rotation trial, participants performed instructed re-aiming trials, with their hands drifting away from the re-aiming target. This implicit drift can be explained equally well by the PPE, PE, and SPE models. **(D)** Results of Experiment 4. Participants exhibited an implicit drift away from the intended direction after instructed re-aiming, with the drift being significantly larger for the 45° re-aiming offset compared to the 90° re-aiming offset. Colored lines with shaded areas represent means and SEMs for the two conditions. **(E)** Extent of implicit learning observed and predicted by the three competing models. The PPE model accurately predicts smaller implicit learning for the 90° condition compared to the 45° condition, while both the PE and SPE models erroneously predict the opposite. Error bars represent SEMs for the data and bootstrap-derived standard deviations for the model. The black dots represent individual data.

**Experiment 5: Performance error based on spontaneous re-aiming fails to induce implicit learning.** In Experiment 4, participants were instructed to re-aim, preventing the spontaneous formation of explicit learning. However, one might argue that PE and SPE could still account for implicit drift if re-aiming occurred spontaneously rather than being explicitly instructed. To address this, Experiment 5 introduced a re-aiming target after participants had fully developed their own explicit learning. In this experiment, participants (n = 20) adapted to a 45° VMR, with explicit learning intermittently measured using aiming reports during adaptation (Fig 7A). After reaching a steady-state learning phase (spontaneous re-aiming in Fig 7B), we presented a re-aiming target that aligned with each participant's last reported aiming direction. Participants were instructed to aim for this re-aiming target for 30 additional trials to assess potential implicit drift (instructed re-aiming in Fig 7B). Since this re-aiming target was consistent with the participants' spontaneous re-aiming direction, both SPE and PPE should remain unchanged (Fig 7C), and both models would predict no further implicit learning. In contrast, the PE model predicts that the introduction of the re-aiming target would create a new performance error ($PE_{re-aiming}$), leading to further implicit learning beyond the learning asymptote (Fig 7C).

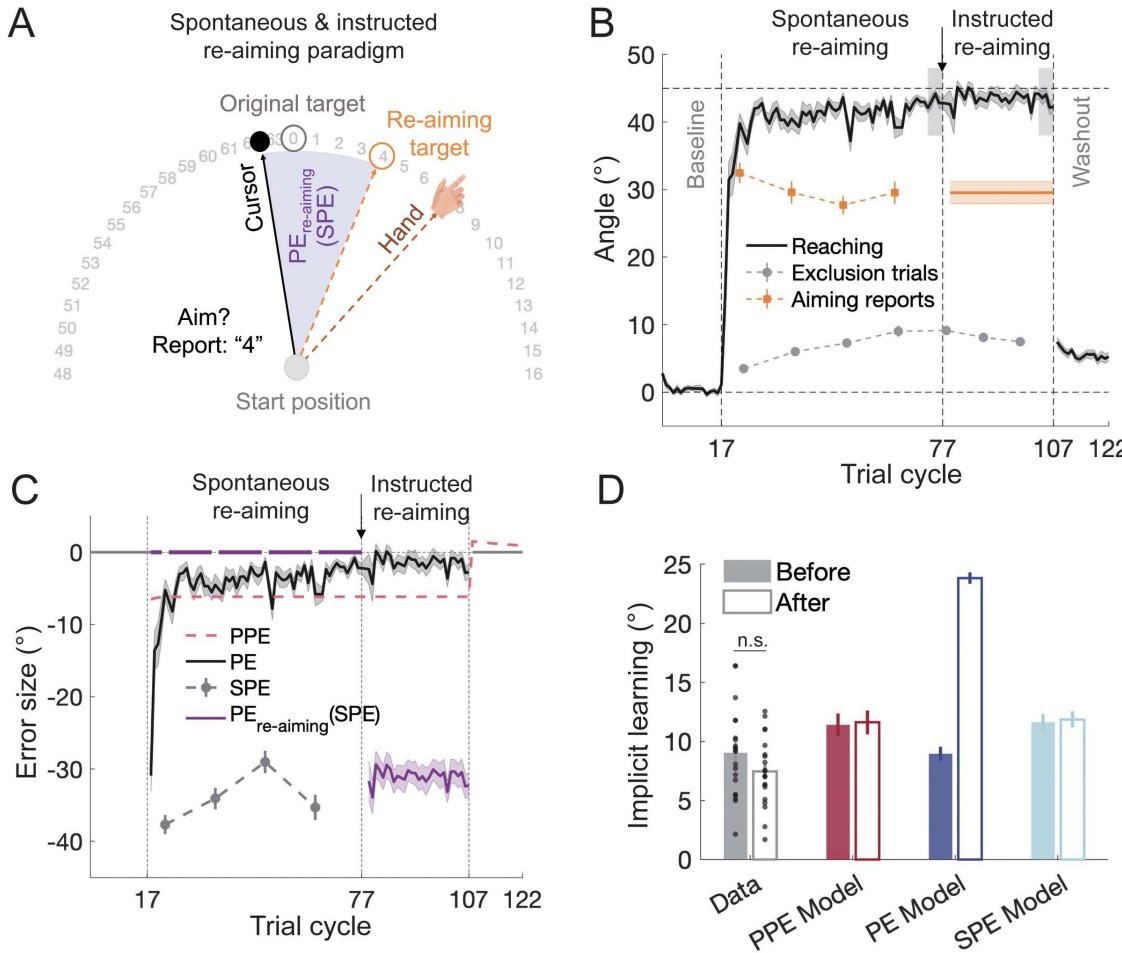

**Fig 7. Results of Experiment 5. (A)** Illustration of the re-aiming paradigm. Participants reported their aiming direction using a numeric ring. After reaching asymptotic learning for a 45° VMR, the last reported aiming direction was used to designate a re-aiming target for each participant. **(B)** Learning in Experiment 5. Participants adapted to a 45° VMR with explicit learning measured through verbal aiming reports and implicit learning assessed using exclusion trials. After reaching the learning asymptote, participants continued to aim at their chosen direction, now marked by a re-aiming target. The introduction of the re-aiming target divides the learning into two phases, one for spontaneous re-aiming and the other for instructed re-aiming. **(C)** Time courses of PPE, PE, SPE, and $PE_{re-aiming}$ before and after the introduction of the re-aiming target. PE, SPE, and $PE_{re-aiming}$ were calculated from the data, while PPE was estimated by model fitting. Note that the introduction of the re-aiming target results in a new $PE_{re-aiming}$, while PPE and SPE remain unchanged. **(D)** Implicit learning observed and predicted by the models. Implicit learning was measured using exclusion trials. The models were fit to the data before the re-aiming target was introduced and used to predict learning afterwards. The PE model failed to predict the observed re-aiming performance, while the PPE model accurately predicted the absence of further implicit learning. Error bars represent SEMs for the data and bootstrapped standard deviations for the model predictions. The black dots represent individual data.

Participants reported an average explicit learning of 29.53°±1.63° in their last aiming report, consistent with previous studies [8,28]. However, no further increase in implicit learning occurred after the re-aiming target was introduced. The hand deviation from the original target with the re-aiming target (43.01°±0.50°) was statistically indistinguishable from the level before introducing the re-aiming target (43.25°±0.79°, $t(19) = 0.327$, $p = 0.747$, $BF = 0.24$). Moreover, implicit learning, as measured by exclusion trials, did not change when comparing the last exclusion trial cycle before re-aiming target introduction to the first trial cycle of the washout phase (9.04°±0.84° vs. 7.44°±0.66°, $t(19) = 1.888$, $p = 0.075$; Fig 7D). These findings show that introducing a re-aiming target after explicit learning has been formed

does not induce further implicit learning, contradicting the PE model's predictions. Instead, the null result aligns with the prediction of the PPE model (Fig 7D), which also provided a better fit to the data ($R^2 = 0.882$ vs. 0.337 for the PE model, S6 Table).

**Experiment 6: Direct measures of PPE reveal its correlation to implicit learning.** Thus far, PPE has been implicated through model analysis or indirectly measured by proprioceptive bias in Experiments 1 and 2. In Experiment 6, we directly measured PPE by asking participants to locate both their aiming and produced movement. Our model simulation indicates that PPE should remain small throughout the learning process (Fig 1C), and should exhibit a similar perturbation size-dependency as proprioceptive bias and implicit learning (Figs 2D and 3D). To test these predictions, we randomly assigned 30 participants to three conditions with 15°, 30°, and 90° VMR, respectively (n = 10 each). For a probe trial, a reaching movement was preceded by a pre-movement aiming report and followed by a post-movement report of perceived hand direction (Fig 8A). The direction difference between these two reports reflects the perceived error. Indeed, when the visual perturbation was turned on, the perceived hand direction deviated away from the aiming direction and towards the visual perturbation (Fig 8B). Thus, the PPE was significantly negative (15°: -1.99° ± 0.84°, $t(9) = -2.36$, $p = 0.042$; 30°: -4.39° ± 0.47°, $t(9) = -9.42$, $p < 0.001$; 90°: -1.86° ± 0.65°, $t(9) = -2.87$, $p = 0.018$), in the opposite direction of implicit learning (Fig 8C). Notably, the size of PPE was rather small and remained small throughout learning, consistent with our initial model simulations (Fig 1C). Importantly, PPE followed a nonlinear concave function of perturbation size ($F(2,27) = 4.52$, $p = 0.020$, post-hoc test: 30° vs. 90°: $p = 0.038$), resembling the model-estimated PPE based on data from Bond 2015 and Experiment 1 (Figs 2D and 3D) and the proprioceptive bias measured in Experiment 1 (Fig 3D). Furthermore, the concave PPE pattern revealed here was accompanied by a similar concave pattern in implicit learning (Fig 8D; $F(2,25) = 5.04$, $p = 0.015$, post-hoc test: 30° vs. 90°: $p = 0.012$). Across participants, PPE also positively correlated with implicit learning (Fig 8E; $R = 0.41$, $p = 0.031$). Hence, the direct measures of PPE provide further support for the driving role of PPE in implicit learning.

## Discussion

Using a combination of systematic behavioral experiments and computational modeling, our study provides converging evidence that explicit and implicit processes in sensorimotor learning are driven by distinct error signals. Explicit learning is driven by performance error, while implicit learning is governed by perceptual prediction error. By leveraging the classical visuomotor rotation paradigm, we disentangled three potential error signals for implicit learning—performance error, sensory prediction error, and perceptual prediction error—across five experiments. Our findings demonstrate that implicit learning exhibits a perturbation size-dependency that can only be explained by PPE, not by PE or SPE (Experiments 1 and 2; [28]). The perceptual prediction error can be indirectly measured by proprioception tests, which show people's perceived hand is biased and this bias is correlated with implicit learning (Experiments 1 and 2). When explicit learning is suppressed using random perturbations, implicit learning follows a nonlinear size-dependency, again consistent only with PPE (Experiment 3). When explicit learning is constrained by instructed re-aiming (Experiment 4), implicit learning develops in a pattern that directly contradicts the predictions of SPE-based learning [15]. When a re-aiming strategy is introduced after explicit learning has already been established, no further implicit learning occurs, refuting the predictions of PE-based learning (Experiment 5). Directly measured by probe trials, PPE is found to remain negative with a small yet significant offset from the aiming direction (Experiment 6), a pattern drastically different from PE and SPE yet matching the model predictions (Fig 1B and 1C). Individual analysis reveals a positive correlation between PPE and implicit learning. Similar to implicit learning and proprioceptive bias, the measured PPE shows a concave dependency on perturbation size, further supporting its role in driving learning and biasing proprioceptive judgment. Overall, these results challenge the prevalent theories that attribute implicit motor learning to either performance or sensory prediction errors. Instead, they highlight the critical role of perceptual prediction error, defined as the deviation of the perceptual estimate of hand motion from the predicted (desired) motion, in driving implicit motor learning.

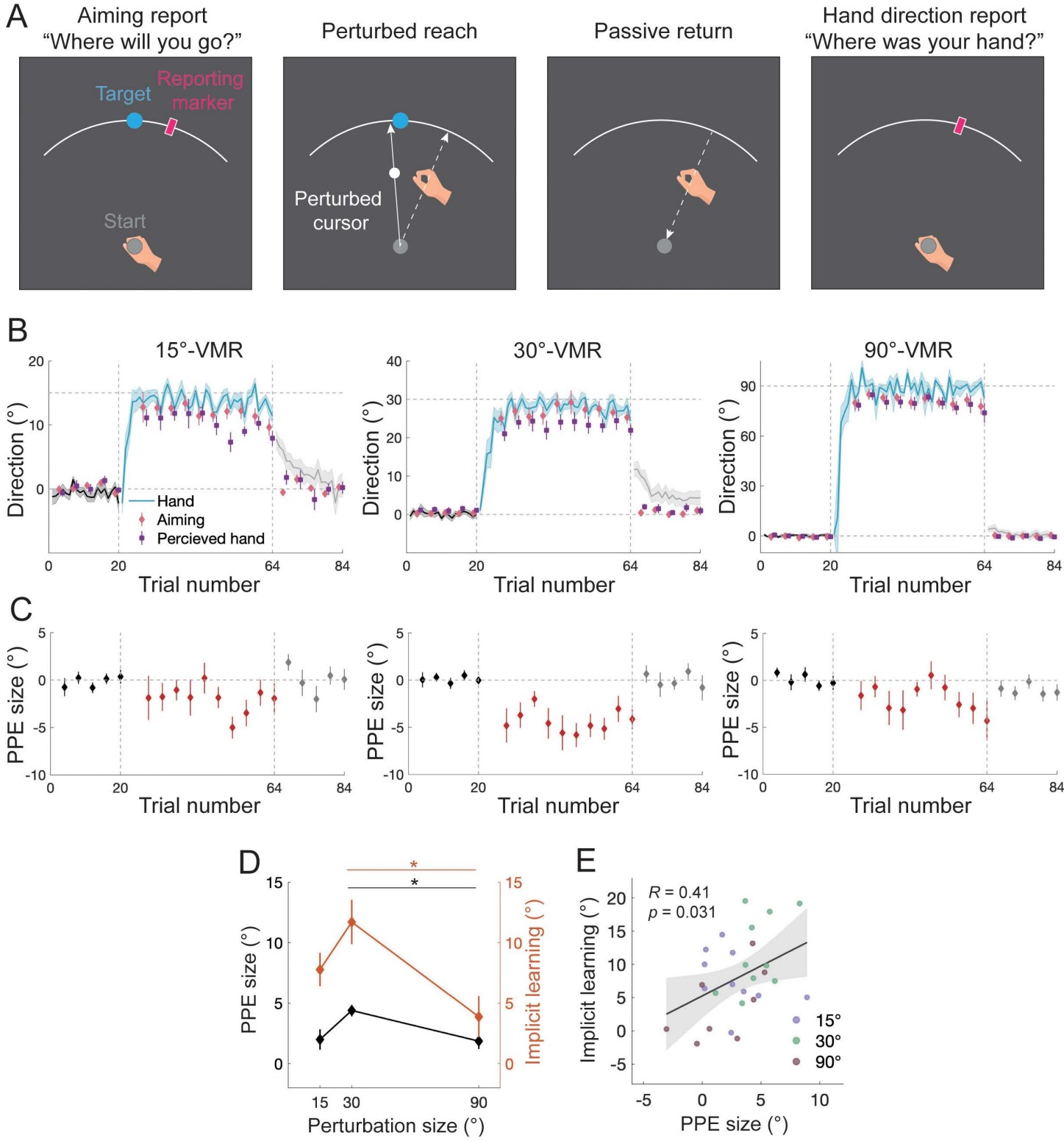

**Fig 8. Results of Experiment 6. (A)** Illustration of the probe trial that directly measures PPE. Participants reported their aiming direction and perceived hand direction before and after a perturbed reach by placing a visual marker with their left hand via the KINARM robot. The just-experienced visual perturbation biases the perceived hand direction away from the aiming direction. **(B)** Learning performance and reported aiming/hand directions from the three perturbation-size conditions are separately shown. Movement direction relative to the target shows typical adaptation changes. Aiming direction

and perceived hand direction change along with learning. Hand direction is more rotated than the aiming direction, manifesting implicit learning; the aiming direction is more rotated than the perceived hand direction, manifesting a perceptual error relative to the intended hand direction (i.e., PPE). Shades and error bars denote SEM. **(C)** PPE as a function of probe trials. PPE becomes negative with perturbation, in line with its causal role in driving adaptation in the positive direction. **(D)** The magnitudes of both PPE and implicit learning show a concave dependency on perturbation size. * denotes $p < 0.05$ for post-hoc comparison. **(E)** Implicit learning significantly correlates with PPE as revealed by generalized linear regression.

## Performance error does not drive implicit motor learning

Performance error is a well-established signal for learning in various domains, including perceptual learning [60], reinforcement learning [61,62], decision-making [63,64], and the explicit components of motor learning [8–10,12,18]. However, whether performance error also underlies the implicit component of motor learning—traditionally associated with procedural learning—remains a topic of heated debate. While the dominant view posits that sensory prediction error drives implicit learning [7,14,15,18,22,23], all the influential state-space models endorse performance error as the key driver [8–10,12]. A major challenge in this debate is the difficulty of dissociating performance error from movement prediction error, as performance error relates to goal attainment and is often tightly coupled with the movement itself. Previous studies have attempted to separate these errors by introducing "target jumps" during movement [23,65–67], thus creating a performance error that is less correlated with the movement. However, such manipulations tend to distract participants, affecting both explicit and implicit learning [23,65,66]. Furthermore, the target jump is likely to be interpreted as an environmental change, which leads to minimal learning within the motor system [59,68]. Indeed, the target jump has been shown to be ineffective in eliciting implicit learning [23,69]. In contrast, our study employed the classical VMR paradigm, which does not require additional manipulations of the perturbation feedback. The presence of explicit re-aiming—whether spontaneously learned or instructed—naturally separates the predicted movement from the goal-seeking cursor, allowing us to dissociate movement prediction error from performance error (Fig 1A). Our findings strongly suggest that performance error cannot account for the various patterns of implicit motor learning observed, despite its clear role in explicit learning.

It is worth noting that Albert and colleagues (2022) also employed a similar modeling approach using extensive datasets from VMR paradigms, but they reached the contrasting conclusion that both implicit and explicit learning are driven by the same performance error. However, their analysis focused exclusively on late learning (final learning extent) without considering the entire adaptation process. In late learning, performance error and perceptual prediction error tend to converge in size (Figs 2B, 3B and 7C), potentially leading to misleading model results. In our study, we demonstrated that early learning—where PE and PPE differ significantly—reveals the limitations of the PE model (Fig 2E). Additionally, Albert et al. did not compare stepwise adaptation with one-step adaptation, a critical contrast that highlights the erroneous predictions of both PE and SPE models (Fig 4D). In summary, by analyzing the full learning process, rather than selective learning segments, our study refutes the notion that performance error drives implicit motor learning.

We also considered the possibility that implicit learning is driven by sensory prediction error and performance error simultaneously. Previous research has suggested that implicit adaptation may comprise a fast and a slow component [70]. We constructed an alternative model where implicit learning's fast and slow components are driven by performance error and sensory prediction error, respectively. However, we found that this two-implicit-state model failed to explain the major findings of the key experiments here (S8 Fig and S8 Table). We want to emphasize that these results do not argue against the possibility that implicit learning contains multiple components with different time scales. Rather, they suggest that the two visual errors, i.e., SPE and PE, are unable to drive the two time-scale components for implicit adaptation. Future studies are needed to examine whether perceptual prediction error identified here can adequately model the potential fast and slow components in implicit adaptation.

## Motor prediction error from multisensory integration, not a single modality

Our findings call for a shift in how prediction error is conceptualized in the context of motor control and learning, emphasizing its perceptual nature rather than a strictly sensory one. Traditional models of motor prediction error have largely focused on visual prediction errors derived from the discrepancy between predicted and actual visual feedback during movements [18,20]. This approach has been applied not only to visual perturbations such as visuomotor rotation and saccadic adaptation [71,72], but also to non-visual perturbations such as force [11,73–75] and inertial perturbations [76,77]. However, accumulating evidence suggests that learning of force and inertial perturbations can occur without visual feedback [78–81], contradicting the idea that visual errors alone drive motor adaptation in these contexts. Defining prediction error solely in terms of visual metrics is increasingly seen as inadequate, as several recent studies have noted [29,33]. Despite this, the field continues to label prediction errors in motor control as sensory prediction errors, reinforcing their unimodal, primarily visual nature [18,20]. Our study provides strong evidence that visual prediction error (SPE) cannot adequately capture the dynamics of implicit learning, even when the perturbation itself is visual. In contrast, perceptual prediction error (PPE), which is derived from multisensory integration, can parsimoniously explain both our data and key findings in the literature [15,28,57]. Notably, PPE and SPE share the same prediction, i.e., both assume that the prediction is about where the *hand* intends to be (aiming direction for VMR). However, they differ in the definition of feedback: SPE uses the visual cursor as feedback to compare with the hand prediction, while PPE uses the perceived hand direction (resulting from multisensory integration) as feedback. Thus, prediction and feedback for SPE do not refer to the same entity (hand vs. cursor), whereas for PPE, they both refer to hand direction. The term "perceptual" in PPE highlights the fact that the feedback used to compute this prediction error is a percept, a posterior estimate of effector localization in Bayesian terms.

Our findings suggest that perceptual prediction error drives implicit learning beyond upper-limb movements, as supported by emerging evidence from other motor learning paradigms. One such example is saccadic adaptation, which explores how the oculomotor system adapts to spatial perturbations of gaze targets [72,82,83]. In this domain, performance error (i.e., the retinal distance between the post-saccadic target and the fovea) does not seem to support implicit learning [72,84]. Similar to reaching studies, saccadic adaptation research has traditionally focused on sensory prediction error (i.e., the deviation between the post-saccadic visual error and an internally predicted visual error [72,84,85]. However, recent modeling work suggests that both motoric and perceptual changes in saccadic adaptation are better explained by a postdiction-based error [86]. This postdiction error refers to the discrepancy between the eye-muscle motor command and the perceived target location, the latter arising from a combination of the actual saccadic error and the predicted saccadic displacement. This concept mirrors our perceptual estimate of hand localization in reaching adaptation, wherein multiple sensorimotor cues are integrated to inform the perceptual error driving adaptation. Notably, saccadic adaptation assumes a stable target location across saccades, reflecting the oculomotor system's assumption of spatial stability [87–89]. Similarly, in reaching tasks, the manual control system assumes that the hand should consistently and accurately reach the intended target. In this light, implicit learning functions to recalibrate the motor system to ensure the maintenance of stability and accuracy, aided by perceptual prediction error.

Perceptual prediction error may also underlie implicit learning in speech adaptation, where the articulatory system adapts to auditory perturbations [90,91]. Most studies in speech adaptation emphasized auditory prediction error, a modality-specific form of sensory prediction error [92–97]. However, recent modeling work has demonstrated that a state estimate of the task, rather than sensory prediction error, best explains the observed learning patterns [98,99]. This state estimate is derived from a combination of the previous articulatory state, an efference copy of the motor command, and auditory feedback, creating a multimodal perceptual estimate that updates the articulatory-to-task transformation—essentially, an internal model. Thus, even in the context of auditory perturbations, a multimodal perceptual estimate, rather than a unimodal auditory prediction error, appears to drive speech adaptation.

 

## Theoretical advance of the PPE model relative to previous work

Tsay and colleagues have recently proposed that implicit adaptation should be understood as a process of proprioceptive re-alignment of the hand towards the target (Proprioceptive re-alignment model, PReMo) [33]. They challenged the traditional visual-centric view and emphasized the critical role of localizing one's effector in motor learning. However, as shown in our previous work [29], the assumption of the PReMo, i.e., the effect of visual perturbation on proprioceptive feedback follows a ramp function of perturbation size, is not supported by empirical findings [29,46]. The PReMo model also predicts that the perceived hand direction should be unbiased in late learning (i.e., re-aligned), contradicting our findings in Experiment 6 which revealed a persistent perceptual error in late learning. In the present study, we further show that PPE is not only applicable for a specific motor learning paradigm, i.e., error-clamp paradigm that invokes implicit adaptation only, but also for more generic motor learning scenario where both explicit learning and implicit co-exist [18].

As stated in the Introduction, the VMR paradigm employed here enabled us to dissociate task goal from aiming direction. Previous works using error-clamp adaptation [29,33] confounded task goal and aiming, preventing us from determining which one serves as motor prediction. The VMR paradigm dissociates participants' aiming (intended movement direction) from the target, thereby enabling dissociation of performance error and sensory prediction error. Nevertheless, our data and model comparisons revealed that neither PE nor SPE can adequately explain error-based implicit learning.

## Dual learning reflects the dual imperatives of motor control

The dual learning based on the two distinct error signals could reflect the dual imperatives of motor control: achieving action goals and resolving sensory conflicts. Researchers have shown that unconscious movements tend to resolve multisensory conflicts, as opposed to achieving explicit goals [100–103]. For instance, in the rubber hand illusion paradigm, participants exhibit subtle, unconscious movements to align visual and proprioceptive cues, not to achieve a specific goal but to reduce sensory conflict [100,101]. This suggests that the motor system, apart from controlling explicit actions, implicitly adjusts movements to minimize sensory mismatches [104,105]. Similarly, in the VMR learning task, sensory feedback from different modalities (e.g., visual and proprioceptive) creates a perceptual error relative to the predicted hand movement, leading the motor system to implicitly recalibrate movements to resolve the conflict. Here explicit learning corresponds to the goal-directed imperative of motor control, driven by PE. In contrast, implicit learning aligns with the imperative of minimizing sensory conflict, driven by PPE.

## The relationship between explicit and implicit learning

Our findings on PPE provide new insights into the relationship between explicit and implicit learning, a central question in motor learning. Previous studies have suggested that these two processes either complement each other [12,34] or compete [10,45,106,107]. However, our PPE model suggests that explicit and implicit learning are functionally independent, driven by distinct error signals, and their relationship is task-specific. In tasks where PE and PPE are in the same direction, explicit and implicit learning can work synergistically, as seen in typical adaptation tasks (Experiments 1 and 2). However, when these errors diverge, the two learning processes can conflict. For example, in Mazzoni & Krakauer (2006) and Experiment 4, when participants were instructed to re-aim explicitly, implicit learning caused a drift away from the re-aiming target, paradoxically worsening performance, i.e., creating a PE in the opposite direction of PPE (Fig 6; clockwise PE and counterclockwise PPE in the illustrated example). This conflict reveals that implicit learning, driven by PPE, can sometimes undermine explicit learning, especially when their errors differ. Only when performance error grows large enough and sustains for an extended period of time do individuals adjust their explicit strategy to correct for performance error [26]. Similar conflict or decoupling between explicit and implicit learning also occurs in the mirror-reversal paradigm, in which implicit learning initially worsens performance [13,108,109], even after a few days of training [108,110,111]. Only after sufficient exposure to this conflict, people slowly develop explicit re-aiming strategies to override implicit learning and

thus improve the overall performance [108,109]. Interestingly, in cases of conflict, implicit learning—driven by the minimization of PPE—tends to be prioritized over performance error correction. This is reminiscent of the competition between Type I (unconscious, fast) and Type II (conscious, slow) cognitive processes in human decision-making [11,112].

Past research has reported that better explicit learners tend to exhibit less implicit learning, implying a competitive relationship between the two processes [10,45,106,107]. However, this conclusion is often based on inter-dependent learning measures, as implicit learning is typically measured using exclusion trials, while explicit learning is inferred by subtracting implicit learning from total learning [10,45,47,55]. Given that most participants reach a learning ceiling, this subtraction method risks producing a false negative correlation. When explicit and implicit learning are measured independently—e.g., using aiming reports for explicit learning and exclusion trials for implicit learning—their relationship does not show a consistent negative correlation but varies depending on the context (S9 Fig). In some cases, there is even a positive correlation between the two [113]. Additionally, during early learning, when learning is not yet complete with a relatively large PE, the negative correlation between explicit and implicit learning disappears (Experiment 5; S9 Fig), challenging the notion that these processes compete for a shared performance error signal.

PPE is defined as the discrepancy between the predicted and perceived hand localization, and the predicted hand localization is determined by explicit learning, i.e., the re-aiming in the case of VMR paradigm. In this sense, explicit learning provides an anchor for deriving the error signal for the implicit learning. This anchoring role of cognitive learning echoes recent findings that the generalization of implicit learning is centered at the re-aiming direction [9,114,115]. In summary, our findings suggest that explicit and implicit learning are independent processes that can interact in a task-specific way. Their relationship is not inherently competitive but rather context-dependent, driven by the alignment or misalignment of the underlying error signals.

## Proprioceptive bias resulted from perceptual bias related to perturbed action

Proprioceptive bias observed in motor adaptation highlights how motor learning can modify perception. Our findings suggest that both motoric and perceptual changes during motor learning might follow the principles of Bayesian cue integration. In implicit motor learning, PPE arises from a misperception of the hand's position ($\hat{x}_{hand}$). Similarly, in proprioception tests, biased judgments result from the same biased cue (provided by previous adaptation trials. Consistent with this model, our results show that both proprioceptive bias and implicit learning are modulated by ongoing adaptation (Figs 3 and 4). The model further explains the varied effects of perturbation size on proprioceptive bias, which can be either linear [50,51] or constant [56], an inconsistency left unexplained. A linear dependency emerges with stepwise perturbations, where each step-up of perturbation incrementally increases the perceived hand position [50,51] (Experiment 2). In contrast, a constant dependency is associated with one-step perturbations, leading to similar across perturbation sizes [56] (Experiment 1). The significant correlation between implicit learning and proprioceptive bias further supports the idea that changes in perceived hand position underlie these perceptual shifts (Experiments 1 and 2). The concept based on PPE model aligns with the view that proprioceptive bias results from multisensory integration [49,52–54]. A few studies have theorized that the internal model formed by motor adaptation underlies proprioceptive bias [54,116,117]. However, the internal model typically develops slowly, but proprioceptive bias develops rapidly with perturbation [49,53,118,119] and diminishes rapidly without it [29]. This makes the internal model an unlikely explanation. Thus, Bayesian cue combination provides a unified explanation for both implicit motor learning and accompanying perceptual changes.

One interesting finding regarding the role of proprioception in implicit adaptation is that the deafferent patients, surprisingly, exhibited normal implicit adaptation [120]. This appears contradictory to our PPE model predictions: the increase in proprioceptive uncertainty among deafferented patients should increase PPE and thus enhance implicit learning if all other factors remain the same. However, deafferented patients undergo lifelong compensatory changes for their proprioceptive loss in both perception and action. For instance, they relied exclusively on efference-based central sense of effort to judge force and weight [121], and they preserved and even exaggerated their anticipatory grip-force scaling and predictive

strategies for object manipulation, while their feedback corrections are impaired [122]. These findings suggested that they re-weight efference-related or predictive cues for sensorimotor control after lifelong compensation. We thus postulate that the observed intact implicit adaptation among deafferented patients might be related to their re-weighting of sensory cues and predictive cues. This hypothesis can be tested by measuring PPE indirectly (using the passive localization tests in our Experiments 1 and 2) or directly (our Experiment 6) among them in future investigations.

## Implications for neurophysiological studies

The compositional nature of motor learning—wherein explicit and implicit learning are driven by distinct error signals—offers a new framework for exploring their neural underpinnings. Traditionally, neurophysiological studies have focused on the neural correlates of total learning, implicating both sensorimotor regions (e.g., cerebellum and sensorimotor cortex [18,123–127], and regions associated with cognitive control and decision-making (e.g., prefrontal cortex and basal ganglia [128–130]). By decomposing total learning into explicit and implicit components, future research can better delineate the neural circuits underlying these distinct processes [131], with particular focus on the cerebellum and posterior parietal cortex (PPC) involved in learning acquisition [22,127,130,132,133]. Furthermore, most previous studies have used either sensory prediction errors [124,134–136] or performance errors [137–140] to identify neural correlates of motor learning. Our finding that perceptual prediction error underlies implicit learning, with a distinct temporal profile from other errors, suggests that more tailored neural assays could be developed [141]. Behavioral paradigms that effectively modulate perceptual prediction error, like the stepwise perturbations in Experiment 2, could be particularly useful for future investigations. Additionally, our results emphasize the importance of localizing one's effector in motor control and learning, requiring integration of motor prediction, visual, and proprioceptive cues on a trial-by-trial basis (Fig 1A). The neural substrates for encoding and integrating these cues overlap with key networks involved in visuomotor learning, including the premotor cortex for intended cursor movement [142,143], PPC for learning-related visual representations [130] and visuomotor transformation [144], the cerebellum for error processing and acquisition of internal models [134–136], and the somatosensory cortex that contributing to the acquisition and retention of motor learning [135–137]. The interplay between motor learning and minimizing position errors of body effectors warrants further examination with neurophysiological approaches. By incorporating insights from our findings, future studies can better understand the neural mechanisms that support the distinct yet interconnected processes of explicit and implicit motor learning.

# Methods

## Ethics statement

The study was approved by the Ethical Committee of the School of Psychological and Cognitive Sciences at Peking University. All participants provided written informed consent to participate in the study and received monetary compensation upon completion of the experiment.

## Participants

We recruited 204 college students from Peking University (118 females, 22.76±2.53 years, mean±SD). All participants were right-handed and had normal or corrected-to-normal vision. Participants were naïve to the purpose of the experiment.

## Apparatus

In Experiments 1 and 2, we used the KINARM planar robotic manipulandum with a virtual-reality system (BKIN Technologies Ltd., Kingston, Canada). Participants sat in a chair and held the two robot handles with their left and right hands, respectively. The right hand performed the reaching task, and the left hand indicated the perceived direction of the right

hand if required. A semi-silvered mirror positioned below eye level displayed visual stimuli and obscured the participants' view of their hands and the robotic manipulandum. The sampling rate of the KINARM was 1,000 Hz.

In Experiments 3, 4, and 5, we used a horizontally placed digitizing tablet (48.8 x 30.5 cm, Intuos 4 PTK-1240, Wacom, Saitama, Japan) to measure people's hand movements. Participants sat in front of a vertically-placed LCD screen (29.6 x 52.7 cm, Dell, Round Rock, TX, US) with a customized wooden blinder placed over the tablet to occlude the vision of the hand. Using a stylus with their right hand, participants performed the reaching task on the tablet. The sampling rate of the tablet was 200Hz. Procedures and data acquisition were custom-coded (Psychtoolbox-3-3.0.19.2 in MATLAB).

The eye-tracking experiment used a similar setup to Experiments 3, 4, and 5, with the addition of eye tracking. A slightly smaller digitizing tablet (31.1 x 21.6 cm, Intuos M, Wacom, Saitama, Japan) was used. The vertically-placed LCD screen was positioned approximately 50 cm in front of a chin rest, with a bracket occluding the participants' view of the tablet and their hands. Movement trajectories were sampled at 200Hz by the digitizing tablet. Eye movements were continuously tracked at 500Hz using a video-based eye tracker (Eyelink 1000, SR Research), which was placed beneath the LCD screen.

**Experiment 1.** Sixty participants were randomly assigned to one of four groups (n = 15 each). Each group adapted to visuomotor rotation of one of four possible sizes (15°, 30°, 60°, or 90°). Participants made reaching movements from a central start position towards an upright (defined as 45°) target located 10 cm away. The start position, target, and cursor were displayed as filled circles, with a 5-mm radius purple circle for the start position, a 5-mm radius blue circle for the target, and a 3-mm radius white circle for the cursor. Each trial began with participants holding their right hand at the start position for 800ms. The target then appeared and signaled participants to make a quick, straight sliding movement through it. The trial ended when the distance between the hand and the start position exceeded 10 cm. If the movement duration from leaving the start position to reaching 10 cm exceeded 300ms, a warning message, "too slow", would appear on the screen.

The experiment included three phases: a 35-trial baseline phase with veridical feedback, a 75-trial adaptation phase with VMR feedback, and a 20-trial washout phase without cursor feedback. The direction of the VMR perturbation was counterbalanced across participants. To assess implicit learning at the asymptote of learning, we inserted exclusion trials at the 50, 55, 60, and 65th trials during the adaptation phase. In these exclusion trials, participants were instructed to move their hand straight to the target without any strategy. The washout phase used the same exclusion instruction, which the experimenter emphasized during data collection.

To measure proprioceptive bias, we inserted proprioception test trials at the 15, 20, 25, 30, and 35th trials during the adaptation phase (Fig 3A). Additional proprioception test trials were conducted at the 5, 10, 15, and 20th trials during the baseline phase to assess baseline proprioceptive bias. In the proprioception test trial, participants were instructed to relax their right arm, which was then passively moved to one of four locations (± 5° and ± 15° away from the reaching target). The passive movement lasted for 1000ms and followed a straight path with a minimum-jerk speed profile. During this movement, a white ring, centered at the start position, expanded in sync with the hand's center-out displacement to indicate the movement distance. After the right hand reached the target location, participants held their right hand steady and used the left robot handle to indicate the perceived position of their right hand. Moving the left handle controlled the rotation of a visual "dial" displayed along an arc that passed through the proprioceptive targets [29]. Participants held the dial at their perceived location for 1,000ms, which was recorded as their perceptual judgement.

**Experiment 2.** Experiment 2 used the same task setup as in Experiment 1. We recruited two groups of participants: one group experienced stepwise perturbations (n = 15), while the other group experienced a one-step perturbation (n = 15). The stepwise group experienced a stepwise 15°→30°→45°→60° VMR perturbation, with each perturbation presented for 40 trials. The one-step group experienced a 60° perturbation for 160 consecutive trials (Fig 4A). Within each 40-trial perturbation block, we inserted exclusion trials at the 30 and 35th trials to measure implicit learning, and proprioception test trials at the 5, 10, 15, and 20th trials to measure proprioceptive bias. Before the adaptation phase, all participants

completed a 35-trial baseline phase with veridical feedback. After the adaptation phase, they went through a washout phase without visual feedback, following the same exclusion instruction.

**Experiment 3.** Experiment 3 investigated the size-dependency of implicit learning when explicit learning was suppressed by the randomness of perturbations. To prevent cumulative learning over trials, three targets (25°, 45°, and 65° at an 8 cm distance) were used. Participants (n = 18) began with a baseline phase consisting of 15 veridical-feedback trials and 15 interleaved no-feedback null trials. In the subsequent adaptation phase, trials were organized into 6-trial mini blocks, where each target was presented once randomly and each perturbation trial was flanked by two no-feedback null trials (Fig 5A). During perturbation trials, participants were instructed to control the cursor to hit the target, while null trials served as exclusion trials. Visuomotor rotation of 0°, 4°, 8°, 16°, 32°, and 64°, either clockwise or counter-clockwise was randomly used in each perturbation trial. These 11 perturbations were repeated nine times, leading to a total of 99 perturbation trials and 99 null trials in the adaptation phase. The wide range of rotation sizes enabled us to uncover the linear or nonlinear size-dependency of implicit adaptation [59,145]. The randomness of perturbation size and direction would prevent participants from forming an adaptive strategy. Single-trial learning was quantified as the difference in movement direction between the two adjacent null trials before and after a perturbation trial. Of note, despite the randomness of the perturbations, we emphasize to participants that the cursor is task-relevant and they need to use the cursor to hit the target despite the perturbation. Although this task-relevant instruction could make the task challenging, it emphasizes the task-relevance of visual feedback, which could affect adaptive learning. A previous study has shown that under these instructions and with random perturbations, participants can fulfil the task without any re-aiming strategy [136].

**Experiment 4.** Experiment 4 examined VMR adaptation with instructed re-aiming, as in the seminar study by *Mazzoni & Krakauer, 2006*. However, two different sizes of VMR were used for two separate groups of participants, i.e., 45° (n = 15) or 90° VMR (n = 15). The target, presented as a 5-mm radius white circle, would appear at one of eight possible locations (0°, 45°, 90°, 135°, 180°, 225°, 270°, and 315°) 8 cm away, randomly in an eight-trial cycle. In the baseline phase, participants were instructed to control the cursor to hit the target with veridical feedback. In the adaptation phase, a new re-aiming target was introduced starting from the first perturbation trial. The re-aiming target, which was either 45° or 90° but in the opposite rotation direction of the imposed VMR, was presented as a 5-mm radius green circle, distinct from the original target. For a consecutive series of 120 trials, participants were required to move their hand to the re-aiming target to make the cursor "hit" the original target. Half of the participants in each group experienced clockwise perturbations and the other half counterclockwise perturbations. Typically, people could maintain their aim at the re-aiming target despite their hand implicitly drifting away from it, across the span of 120 trials [26].

**Experiment 5.** Experiment 5 examined the instructed re-aiming introduced at the end of VMR adaptation, as opposed to at its beginning (as in Experiment 4). Four targets were used, located at 45°, 135°, 225°, and 315°, 8 cm away, with trials organized into 4-trial cycles. The baseline phase consisted of 3 trial cycles with veridical feedback trials and 1 trial cycle without feedback. The subsequent adaptation phase consisted of 60 cycles with a 45° VMR. Within the adaptation phase, we intermittently measured the explicit adaptation through aiming reports at the 5, 19, 33, and 47th cycles. In these trials, the target was surrounded by a ring of 63 numbered visual landmarks spaced 5.625° apart, with 0° centered on the target. Before the reach, participants verbally reported the landmark number they planned to reach. Their reported aiming directions were recorded by the experimenter. To measure implicit adaptation, we inserted exclusion trials with no visual feedback at the 6, 20, 34, 48th cycles. In the ensuing re-aiming phase, we presented a re-aiming target at the last reported aiming direction, tailored to each participant. Participants were required to move their hand straight to this re-aiming target for 40 cycles. Note that the re-aiming target would still make the cursor reach the original target since it was aligned with the participant's spontaneous explicit learning asymptote. We inserted exclusion trials at the 1st, 11, and 21st cycles in the re-aiming phase. In the final washout phase, participants completed an additional 15 cycles of exclusion trials.

**Experiment 6.** Experiment 6 aimed to directly measure PPE, defined as the difference between the intended movement direction (aiming direction) and the perceived hand direction. Using a similar setup and procedures as Experiments 1 and 2, Experiment 6 recruited thirty participants and randomly assigned them to learn one of three VMR perturbations (15°, 30°, and 90° VMR; n = 10 each). Participants went through three learning phases, i.e., baseline (20 trials), adaptation (44 trials), and washout (20 trials). Every four trials, they performed a probe trial for measuring PPE (Fig 8A). The first eight adaptation trials were not probed to avoid affecting participants' formation of strategic re-aiming during this critical initial learning phase [146]. For each probe trial, participants held their right hand at the start position for 800 ms, and then a target (a 5-mm radius blue circle), a white arc, and a purple visual marker were displayed. Participants were asked to report their intended aiming direction ("Where will your right hand go?") by moving the left KINARM handle to dial the visual marker, just as in Experiment 2. Participants subsequently performed the reaching movement, with or without the cursor perturbation, by their right hand. After the reach, their right hand was passively returned to the starting position by the robotic manipulandum. The arc and visual marker reappeared to prompt participants to report the perceived direction of their actual hand movement ("Where was your right hand?"), again using the left KINARM handle to dial the visual marker. PPE was computed as the angular difference between these two reported directions.

## Eye-tracking experiment

The eye-tracking experiment examined whether people fixate in the aiming direction during reaching movements in typical VMR adaptation. Participants (n = 16) learned 45° and 90° VMR perturbations in two consecutive sessions, with the order counterbalanced. Each session included a baseline, adaptation, and washout phase, as in Experiment 1, with 40, 160, and 20 trials, respectively. Each trial began when the participant moved the cursor (a 3-mm radius white circle) into the starting position (a 5-mm radius purple circle) using a stylus. After holding the position for 500ms, a target (a 5-mm blue circle) was presented 10 cm away at a random direction between 0° and 360°. Participants started to reach the target with the cursor once the target disappeared. The target presentation duration followed a uniform distribution from 800 to 1200 ms, making the movement trigger less predictable. If the movement was initiated before the target disappearance, the trial was aborted and a text message "reacting too fast" would appear on the screen. Veridical or perturbed cursor feedback was shown during the movement and the target reappeared once the movement ended and lasted for 1000 ms. The disappearance of the target prevented some participants from keeping their fixation at the starting position without a change to their aiming direction. To avoid slow movements, a text message "moving too slow" would appear if the movement time exceeded 400ms.

## The three error models

*Perceptual prediction error (PPE) model* The PPE model hinges on the perceptual estimation of effector position that is dynamically updated through Bayesian cue combination. For reaching, the perceptual estimate of the hand movement direction ($\hat{x}_{hand}$) results from the integration of three sensory cues: the visual cue ($x_v$) from the cursor, the proprioceptive cue ($x_p$) from the hand, and the sensory prediction of the reaching action ($x_u$). Following Bayesian principles [43], the perceptual estimate is

$$\hat{x}_{hand} = \sum_i W_i x_i, \quad with \quad W_i = \frac{\frac{1}{\sigma_i^2}}{\sum_j \frac{1}{\sigma_j^2}}, \quad i, j = u, p, v$$

(1)

where the weight of each cue is determined by its uncertainty ($\sigma$) relative to each other. Notably, in the VMR paradigm, the predicted reaching direction is the aiming direction. The rotated cursor ($x_v$) biases the hand estimate ($\hat{x}_{hand}$) toward the cursor's direction. The deviation from the predicted (aiming) direction constitutes the perceptual prediction error, which

drives implicit adaptation on the subsequent trial (Eq. 2). We assume that learning proceeds on a trial-by-trial basis thus a state-space framework is used. Explicit learning is driven by performance error as in existing models (Eq. 3; 8, 9, 10, 40, 147). For each trial, the learning is updated by

$$x_{t+1}^i = A^i x_t^i - B^i \left( \hat{x}_{Hand,t} - x_{u,t} \right) \tag{2}$$

$$x_{t+1}^e = A^e x_t^e - B^e e_t^{PE} \tag{3}$$

$$x_{t+1} = x_{t+1}^e + x_{t+1}^i \tag{4}$$

Where total learning ($x$) is the sum of explicit learning ($x^e$) and implicit learning ($x^i$). $A$ is the retention rate capturing the inter-trial forgetting and $B$ is the learning rate capturing the proportion of error correction.

As individuals fixate near the aiming direction during the movement [41,42], the visual cursor directionally differs from the fixation by an amount of ($x_v - x_u$). This eccentricity linearly increases visual uncertainty about cursor motion direction, as shown both in visual perception studies and in our previous work [29]. Thus, the visual uncertainty ($\sigma_v$) follows a linear function:

$$\sigma_v = k * \left| x_v - x_u \right| + b \tag{5}$$

where $k$ and $b$ are the slope and intercept of the visual uncertainty, respectively.

*Performance error (PE) model* The performance error (PE) model, also called the competition model [10] or the two-state model [11], has been widely used to explain motor adaptation [8–13]. The Eq. 2 in the PPE model is replaced by

$$x_{t+1}^i = A^i x_t^i - B^i e_t^{PE} \tag{6}$$

*Sensory-prediction error (SPE) model* Implicit motor learning has been suggested to be driven by sensory prediction error [15,22,31]. In the VMR paradigm, SPE is defined as the deviation of the perturbing cursor from the aiming direction [20]. Therefore, the SPE model replaces Eq. 2 in the PPE model by

$$x_{t+1}^i = A^i x_t^i - B^i \left( x_t^i - r \right) , \tag{7}$$

where ($x_t^i - r$) is SPE and $r$ is the VMR size.

## Data analysis

*Hand kinematics:* In Experiments 1 and 2, hand positions and velocities were directly acquired from the KINARM robot at a fixed sampling rate of 1,000 Hz. The raw kinematic data were smoothed using a fifth-order Savitzky-Golay filter with a window length of 50 ms. In Experiments 3, 4, and 5, hand positions were recorded at a sampling rate of 125 Hz. The movement direction was determined by computing the direction of the vector spanning from the starting position to the hand position at the peak outward velocity. We defined the direction to counter the perturbation as the positive direction.

*Gaze:* The gaze position obtained in the eye-tracking experiment was first preprocessed to remove blinks. Consistent with previous studies [41], we used the last fixation before movement onset to quantify the participants' aiming direction. Trials were excluded if the fixation before movement onset was missing (due to a blink), or near the start position (< 40% of target distance), or beyond the target (>150% of target distance). These exclusions amounted to 1.67% of the trials

in the 45°-VMR condition and 3.61% in the 90°-VMR condition. Other direction measures, such as the gaze position at movement onset, yielded similar results.

*Statistical analysis:* For the dataset from *Bond & Taylor 2015,* we quantified early and late learning by using the data from the first 5 cycles and the last 5 cycles in the adaptation phase, respectively. Total learning was defined as the hand's directional deviation from the target. Explicit learning was quantified as the direction deviation of the aiming report from the target. Implicit learning was calculated by subtracting explicit learning from total learning. Group differences were examined using one-way independent-measure ANOVAs. In Experiments 1 and 2, implicit learning was quantified as the average hand deviation from the target in the four exclusion trials. Total learning was calculated as the average hand deviation from the target in the last cycle during the adaptation phase. Explicit learning was calculated by subtracting implicit learning from total learning. Proprioceptive bias was quantified as the average angular difference between the perceived (reported) and actual hand directions over the six proprioception test trials. Group differences were examined using one-way independent-measure ANOVAs, with pairwise post-hoc comparisons conducted using Bonferroni correction. For Experiment 2, two-way mixed ANOVAs were performed, with group as the between-group factor and learning time as the within-group factor. For the correlation analysis of Experiment 2, we used a mixed linear model: $y = \beta_0 + \beta_1 * x_{prop} + \beta_2 * x_{sub|prop} + \epsilon$, where $y$ is implicit learning, $x_{prop}$ is proprioceptive bias, $x_{sub|prop}$ is the random factor of subject, and $\epsilon$ is the residual error. In Experiment 3, single-trial learning was defined as the angular difference in movement direction between two consecutive no-feedback null trials before and after a perturbation trial. Given the symmetry between CW and CCW perturbations, we pool the data to quantify the size dependency on the absolute perturbation size. Repeated-measures ANOVAs were used to examine the effect of perturbation size on single-trial learning. In Experiment 4, independent t-tests were used to compare implicit learning between 45°-VMR and 90°-VMR conditions. In Experiment 5, we used paired t-tests to compare implicit learning before and after introducing the re-aiming target. Implicit learning was quantified by both subtraction and exclusion methods. In Experiment 6, we used one-sample *t*-tests to examine whether PPE was significantly negative, and used one-way ANOVAs to examine group differences in PPE and implicit learning. Post-hoc comparisons were conducted with Bonferroni correction. In the eye-tracking experiment, we used paired *t* tests to compare condition differences (90° vs. 45°) in the re-aiming direction. We used a mixed linear model for analyzing the correlation between explicit learning and eye-fixation direction: $y = \beta_0 + \beta_1 * x_{aim} + \beta_2 * x_{condition} + \beta_3 * x_{sub|(aim*condition)} + \epsilon$, where $y$ is explicit learning, $x_{aim}$ is eye-fixation direction, $x_{sub|(aim*condition)}$ is the random factor of subject, and $\epsilon$ is the residual error. For all experiments, we excluded the trials with movement time exceeding 400 ms or below 150 ms, as well as the trials with systematic movements to the incorrect direction (i.e., movement direction less than -20° or exceeding 120°). These criteria resulted in the removal of less than 1% of trials on average for each experiment.

## Model analysis

Both PE and SPE models have four parameters $\Theta = [A^e, B^e, A^i, B^i]$. The PPE model has one additional parameter: the slope of visual uncertainty as a function of visual eccentricity ($k$). Other parameters related to sensory uncertainty ($\sigma_u, \sigma_p$, $b$) are fixed parameters derived from our previous work using the error-clamp paradigm [29] ($\sigma_u = 5.05°$, $\sigma_p = 11.12°$, $b = 1.853°$). These parameters represent systematic properties of the sensorimotor system that are not task-dependent and thus should remain constant across paradigms. But we left $k$ as a free parameter because the previous work used the error-clamp paradigm, which reinforced fixation on the reach target rather than on the cursor. In contrast, participants in the VMR paradigm here, look at the cursor at the end of the reaching movement, which presumably reduces the effect of eccentricity on visual uncertainty. Not all the experiments need the full models (as detailed below).

*Bond & Taylor, 2015.* We fitted the trial-by-trial average implicit learning data from all the four perturbation conditions simultaneously (Fig 2C). The PPE model had three free parameters $\Theta = [k, A^i, B^i]$, while the PE and SPE models each had two free parameters $\Theta = [A^i, B^i]$. Bootstrapping was used to estimate the standard deviations of the model fits.

Participants in each group were resampled with replacement 5000 times, each time producing a new average learning sequence.

*Experiments 1 and 2.* The trial-by-trial learning data include total learning and implicit learning (intermittently measured during the adaptation phase) from all the four conditions (Experiment 1; Fig 3C) or the two conditions (Experiment 2; Fig 4D). The PPE model had five free parameters $\Theta = [k, A^i, B^i, A^e, B^e]$. Since the one-step condition in Experiment 2 was identical to the conditions in Experiment 1 (only differing in the duration of the adaptation phase), we applied the $k$ value estimated from Experiment 1 to the one-step condition in Experiment 2. The stepwise condition, however, retained $k$ as a free parameter ($k_{step}$ in S3 Table) since the participants would be attracted by the sudden stepwise change in cursor rotation, which would reduce visual uncertainty. Through fitting the data from both two conditions (Fig 4D), we obtained the value of $k_{step}$. We used the values of $k$ and $k_{step}$ for cross-validation (S2B Fig). The SPE and PE models had four free parameters $\Theta = [A^e, B^e, A^i, B^i]$.

*Experiments 3, 4, and 5.* We first assumed that the single-trial learning in Experiment 3 was implicit. Thus, the PPE model had three free parameters $\Theta = [k, A^i, B^i]$ to fit the average trial-by-trial changes (Fig 5B). The averaging was feasible since the pseudo-random perturbation sequence was identical for each participant. For fitting, we set the lower limit of $A^i$ at 0.7 for all the models. Based on fitting the trial-by-trial learning, we calculated the model-predicted single-trial learning as a function of perturbation size (Fig 5D). To examine whether there was explicit learning at work, we then included the explicit learning component and evaluated whether it improved model fitting. In Experiment 4, there was only implicit learning since participants were unaware of their hand drifting away from the instructed re-aiming target. Thus, only implicit learning was fitted for the 45°- and 90°-VMR conditions at once. The PPE model had three free parameters $\Theta = [k, A^i, B^i]$, while the SPE and PE models had two free parameters $\Theta = [A^i, B^i]$. For Experiment 5, the data used for model fitting included the trial-by-trial changes in total learning as well as the intermittently-measured explicit learning. Models were fitted to the data before introducing the re-aiming target and then used to predict the learning afterwards (Fig 7D). The PPE model thus had five free parameters $\Theta = [k, A^i, B^i, A^e, B^e]$, while the other two models had four free parameters $\Theta = [A^i, B^i, A^e, B^e]$.

All model fittings were conducted using *fmincon* function in MATLAB (2022b, MathWorks, Natick, MA, US). To ensure the model adhered to established assumptions, we constrained the parameters with $A^e < A^i$ and $B^e > B^i$ as commonly assumed for explicit and implicit learning components [8,9,11]. All the obtained parameter values, along with quality measures of the model fit, including $R^2$, root mean squared error (*RMSE*), Akaike information criterion (AIC), and Bayesian information criterion (BIC), were provided in S1 to S6 Tables.

## Supporting information

**S1 Fig. The eye-tracking experiment examining the eye fixation pattern during visuomotor rotation adaptation.** (A) Illustration of the experimental procedure. In each trial, participants first held their hand at the start position for 500 ms, awaiting the appearance of a target randomly positioned between 0° and 360°. They were instructed not to move until the target disappeared (~1000 ms after appearance). Visual feedback of the hand cursor was withheld until the trial ended, when it re-appeared together with the target as performance feedback. Eye tracking measurements were taken during reaching. (B) Time course of the gaze direction. Different colors represent different timings of a trial, including target preview, reaction time (RT), movement duration, and terminal feedback. Two representative trials from a participant are shown for the 45°-VMR (upper panel) and 90°-VMR (lower panel) conditions, respectively. The participant fixated on the starting position during the target preview and RT, then shifted fixation toward the re-aiming direction during the movement, maintaining this fixation until the terminal feedback was provided (purple line). The re-aiming direction was less rotated than the imposed VMR. After the movement ended and visual feedback was provided, participants tended to fixate back to the cursor (yellow line). We calculated the probability of fixations in the target area (± 22.5° around the target direction) after

movement: 64.28±4.91% in the 45°-VMR condition and 43.23±6.80% in the 90°-VMR condition. (C) Average hand movement and eye fixation directions during adaptation to the 45° and 90° VMR. The fixation direction was defined as the last fixation before movement onset [41]. The shaded areas denote the standard errors across participants. From early learning, participants started to fixate away from the target direction, closer to their explicit re-aiming direction, and this fixation "strategy" persisted throughout the adaptation phase. Note that the fixation deviated further from the target in the 90°-VMR condition compared to the 45°-VMR condition, consistent with the condition difference in re-aiming and explicit learning. In the washout phase, their fixation returned to the target as instructed. (D) Participants' fixation direction in the 90°-VMR condition was significantly higher than that in the 45°-VMR condition, during both early (45°: 19.00°±2.23°; 90°: 44.49°±4.43°; $t_{(15)}$ = -6.38, $p < .0001$) and late learning (45°: 21.63°±2.56°; 90°: 57.88°±4.18°; $t_{(15)}$ = -9.21, $p < .0001$). The larger the fixation direction, the greater the re-aiming deviation away from the target. Early and late learning refer to the first 5 and last 5 trial cycles during adaptation. (E) Participants' eye-fixation direction was positively correlated with their explicit learning ($\beta$ = 0.80, $p = 0.014$), the latter estimated from the first exclusion trial in the washout phase.
(TIF)

**S2 Fig. The performance of the PPE model was compared by assuming a linear and an exponential visual uncertainty function.** The observed and model-predicted implicit learning for Bond & Taylor 2015 (A), Experiment 1 (B), and Experiment 2 (C). The two versions of the PPE model similarly explain the data. (D) The visual uncertainty functions are estimated from the three datasets. Note the estimates are similar for the linear and the exponential functions, and they are also close to the visual uncertainty measured in our previous work [29].
(TIF)

**S3 Fig. Data and model fitting results for total and explicit learning in Experiment 1.** All three models reproduced the scaling effect of perturbation size for total and explicit learning. Error bars represent SEM for data and bootstrapped standard deviations for the model. The black dots represent individual data.
(TIF)

**S4 Fig. To demonstrate the reliability and robustness of our modeling approach, we conducted parameter recovery analysis for the PPE model.** We chose Exp.1's dataset to show the case when the model fits both explicit and implicit learning, and Exp.4's to show the case when the model only fits implicit learning, as explicit learning is constrained by re-aiming instructions. For each experiment, we employed a bootstrap resampling approach (N = 5,000) to estimate the parameter distributions. We then used the median±standard deviation of the bootstrapped parameters to define realistic parameter ranges for simulation. Critically, we incorporated motor noise into the simulations, estimated from the baseline movement variability of participants to ensure that our synthetic data reflected realistic noise characteristics. X-axis shows simulated parameters and y-axis shows recovered parameters. Simulated and recovered parameters showed high concordance, with concordance correlation coefficient (CCC) exceeding 0.98 for all three parameters in Exp.1 (A), and strong concordance with CCC exceeding 0.8 in Exp.4 (B). These results show that our fitting pipeline are reliable under realistic noise and sample sizes.
(TIF)

**S5 Fig. To demonstrate the reliability and robustness of our modeling approach, we performed confusion matrix analysis for model discriminability.** Confusion matrices of the three competing models (PPE, PE, and SPE models), using datasets from Exp.1 and Exp.4. We generated synthetic data from each generative model and applied our model selection procedure (the model with the lowest BIC) to these simulated datasets to assess whether the generative model could be correctly identified. The confusion matrices display the proportion of simulations in which each true generative model was recovered as the selected model. The results show strong discriminability among the three models.
(TIF)

**S6 Fig. More model comparison results from Experiment 2, using variable explicit learning and cross-validation.** (A) Implicit learning data in Experiment 2, a reproduction of Fig 4C right panel. (B) Model fitting with varying explicit parameters across the two conditions. Compared to the model fitting in the main text (Fig 4D), here we allowed explicit parameters to vary across the two conditions. The PE and SPE models still failed to reproduce the data, especially the relative size of implicit learning between the two perturbation conditions. The PE model could not capture the implicit learning patterns for small stepwise perturbations (15° and 30°), and it yielded unrealistic model parameters (Table B in S3 Table). The SPE model showed minimal improvement, continuing to underestimate implicit learning in the stepwise condition and overestimate it in the one-step condition. The PPE model remained capable of explaining the data well with variable explicit learning parameters. These results suggest that the failure of PE and SPE to explain implicit learning across conditions was not due to an unconsidered potential parametric change in explicit learning. (C) Cross-validation tests for the three models. Each model was fit to the data from the stepwise condition and then used to predict implicit learning in the one-step condition. Neither PE nor SPE model captures the learning pattern: their predictions for the one-step group were highly inaccurate; both models overestimated implicit learning and underestimated explicit learning in the one-step condition. In contrast, the PPE model's predictions match well with the empirical data across conditions, further reinforcing its robustness in capturing the dynamics of both implicit and explicit learning across different perturbation schedules. In (B) and (C), error bars represent bootstrapped standard deviations (resample size = 5,000). (TIF)

**S7 Fig. Estimated values of the parameter *k* (the slope of visual uncertainty as a function of visual eccentricity) of the PPE model across different experiments [28].** Pink diamond markers indicate median values estimated using bootstrap resampling (N = 5,000), with horizontal error bars representing 95% confidence intervals. Purple line markers represent parameter values estimated by fitting the average learning data, which closely align with the bootstrap median estimates, demonstrating robust and consistent model fits across experiments. (TIF)

**S8 Fig. Model analysis using a two-state model for implicit learning, which assumes that implicit learning comprises fast and slow components driven by both performance error (PE) and sensory prediction error (SPE), respectively.** We used different datasets to test the model: (A) Bond & Taylor, 2015; (B-D) Our Experiments 1, 2, and 4. The model analysis showed that the two-state model failed to explain the data. Furthermore, parameter estimates (a very small *B*) from the model fitting suggest that incorporating an additional implicit learning component is unnecessary (S8 Table). (TIF)

**S9 Fig. Correlation between explicit and implicit learning varies widely when the two learning components are independently measured.** Data from three separate studies are presented, with explicit learning measured by aiming reports and implicit learning measured by no-feedback exclusion trials. (A) Correlations between explicit and implicit learning in different perturbation size conditions, including 15°, 30°, 60°, and 90° VMR (n = 10 for each). Due to the small sample size and variability across conditions, support for a negative correlation was insufficient. Data from *Bond & Taylor, 2015*. (B) Correlations across perturbation size conditions of 15°, 30°, 45°, and 60° VMR (n = 33, 54, 28, 61, respectively) from *D'Amario et al., 2024*. With decent sample sizes, positive, negative, or null correlation was observed in these different conditions. (C) Correlations were inconsistent when the format of visual perturbation varied, but with a fixed perturbation size of 45° VMR. Data from *D'Amario et al., 2024*. Continuous cursor feedback (n = 51) is the conventional feedback format, terminal feedback promotes explicit learning and reduces implicit learning (n = 35), delayed feedback reduces implicit learning (n = 39), and cursor jump feedback usually promotes explicit learning and reduces implicit learning (n = 32). Only the continuous-feedback condition showed a negative correlation; no correlation was detected for the other three feedback formats. (D) Early learning did not show a negative correlation. In our Experiment 5 (n = 20), explicit

learning and implicit learning were independently measured four times during the 60-trial-cycle learning period. The two early instances (the 5th and 19th cycles) did not show significant negative correlations, the third instance (the 33rd cycle) showed a marginal effect, and the fourth instance (the 47th cycle) showed a significant negative correlation. Thus, though performance error is most prominent during early learning, there is no evidence that the two learning processes compete for this error and then lead to a negative correlation. Each dot represents an individual's data. (TIF)

**S1 Table. Model comparisons and parameters for the dataset from *Bond & Taylor, 2015.***
(XLSX)

**S2 Table. Model comparisons and parameters for the data from Experiment 1.**
(XLSX)

**S3 Table. Model comparisons and parameters for the data from Experiment 2.**
(XLSX)

**S4 Table. Model comparisons and parameters for the data from Experiment 3.**
(XLSX)

**S5 Table. Model comparisons and parameters for the data from Experiment 4.**
(XLSX)

**S6 Table. Model comparisons and parameters for the data from Experiment 5: Cross-validation (Fitting before-re-aiming phase, predicting after-re-aiming phase).**
(XLSX)

**S7 Table. Model comparisons of the PPE models, which assume linear, exponential, or power-law visual uncertainty function.**
(XLSX)

**S8 Table. Model analysis using a two-implicit-state model, which assumes that implicit learning comprises fast and slow components driven by both performance error (PE) and sensory prediction error (SPE), respectively.**
(XLSX)

## Acknowledgments

We would like to thank Dr. Jordan Taylor for providing the raw data from his study [28].

## Author contributions

**Conceptualization:** Xiaoyue Zhang, Kunlin Wei.

**Data curation:** Xiaoyue Zhang, Wencheng Wu.

**Formal analysis:** Xiaoyue Zhang.

**Funding acquisition:** Kunlin Wei.

**Investigation:** Xiaoyue Zhang.

**Methodology:** Xiaoyue Zhang, Wencheng Wu, Kunlin Wei.

**Software:** Xiaoyue Zhang, Wencheng Wu.

**Supervision:** Kunlin Wei.

**Validation:** Xiaoyue Zhang, Kunlin Wei.

**Visualization:** Xiaoyue Zhang.

**Writing – original draft:** Xiaoyue Zhang, Kunlin Wei.

**Writing – review & editing:** Xiaoyue Zhang, Kunlin Wei.

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
