## [Decision Letter · Decision Letter 0]

5 Nov 2025

Perceptual prediction error supports implicit process in motor learning.

PLOS Computational Biology

Dear Dr. Wei,

Thank you for submitting your manuscript to PLOS Computational Biology. After careful consideration, we feel that it has merit but does not fully meet PLOS Computational Biology’s publication criteria as it currently stands. Therefore, we invite you to submit a revised version of the manuscript that addresses the points raised during the review process.

We look forward to receiving your revised manuscript.

Kind regards,

Tianming Yang

Academic Editor

PLOS Computational Biology

Andrea E. Martin

Section Editor

PLOS Computational Biology

**Journal Requirements:**

At this stage, the following Authors/Authors require contributions: Kunlin Wei. Please ensure that the full contributions of each author are acknowledged in the "Add/Edit/Remove Authors" section of our submission form.

5) We notice that your supplementary Figures, and Tables are included in the manuscript file. Please remove them and upload them with the file type 'Supporting Information'. Please ensure that each Supporting Information file has a legend listed in the manuscript after the references list.

Potential Copyright Issues:

i) Figures 1A, 5A, 6A, 6B, 7A, 8A, and S1A. Please confirm whether you drew the images / clip-art within the figure panels by hand. If you did not draw the images, please provide (a) a link to the source of the images or icons and their license / terms of use; or (b) written permission from the copyright holder to publish the images or icons under our CC BY 4.0 license. Alternatively, you may replace the images with open source alternatives. See these open source resources you may use to replace images / clip-art:

7) Please amend your detailed Financial Disclosure statement. This is published with the article. It must therefore be completed in full sentences and contain the exact wording you wish to be published.

**Reviewers' comments:**

Reviewer's Responses to Questions

Reviewer #1: In this paper, Zhang et al. continue to explore the recent perceptual prediction error ("PPE") model of implicit motor adaptation. To do this, they both re-analyze previously published datasets, and also performed several new experiments to pit the PPE model against other prominent ideas about the errors signals that drive motor learning, the sensory prediction error (SPE) idea, and the performance error (PE) idea. Their experiments involve novel behavioral manipulations, as well as computational modeling, proprioceptive assays, and eye-tracking. Overall, the authors argue that the data best support the PPE model of motor adaptation, which involves a Bayesian weighting of proprioception and vision to estimate hand position given a desired movement and compare it to observations, and argues strongly against the SPE and PE models.

Even though the model presented here has been previously outlined, this study represents a rigorous and expanded testing of that model in a series of novel and impressive experiments and modeling analyses. In general I read this article with great enthusiasm --- by the end of it, I was rather convinced that the authors' PPE model is indeed the current best model of implicit motor adaptation. My critiques are mostly minor. I applaud the authors on a very thorough and rigorous set of experiments.

Strengths:

- the study uses models to make clear predictions, and clever experiments to test those predictions. overall, the results are rather impressive and lend rather strong support to the PPE model

- the figures are clear and nicely detailed

- the methods are clear and the range of control analyses really adds to the robustness of the modeling

- the eye-tracking experiment is a very nice touch

- the broader implications of the study are nicely laid out in the Discussion, especially w/r/t other forms of motor learning (e.g., speech).

Weaknesses:

- i was somewhat confused about the implied framing of SPE as not being "perceptual"; it seems that an SPE model can easily assume that the visual error is routed through the brain's perceptual computations, not just some kind of direct sensory readout (whatever that would be anyway). the nomenclature used thus makes the nature of the SPE model seem somewhat odd. isn't the main advance here that the prediction errors that drive implicit motor learning involve an uncertainty-weighted perceptual estimate of the hand, not just the visual error? Isn't that what primarily differentiates SPE and PPE, rather than "sensation versus perception"?

- there could be more caveats offered for why the PPE model fit in Figure 2C is not particularly strong. i understand that it clearly outperforms the SPE and PE models; but why is it not a better fit here? this is a small point as the fit is better in the later experiments, but still warrants some discussion.

- there are some data with rather low asymptotic implicit learning (~10 degrees or so). why is this the case? considerations of experimental apparatus lag, or temporally volatile adaptation, could be offered to address this (see recent papers by Hadjiosif et al on these two issues). could it be that there are different aspects of implicit adaptation (e.g., volatile and stable) that have different error sources?

- as a discussion point, i'm curious what the authors think about the fact that deafferented patients can show somewhat normal implicit adaptation (Tsay et al., 2024). this seems like an interesting mystery.

Minor points:

- the legend at the top right of figure 4B has incorrect color labels

- the dark versus light gray shadings in figure 4A are hard to distinguish

- the instructions noted in figure 5 to "make the cursor hit the target" are confusing; don't the subjects not yet know the rotation sign/magnitude on such trials? how would they interpret that instruction?

- it would be nice to see a histogram of k values across all experiments, or bar graphs for each experiment.

Reviewer #2: This manuscript reports six experiments that bolster a previously proposed perceptual prediction error (PPE) account of motor learning. The data are solid and the writing is clear. The replication across paradigms is valuable.

That said, I do not see a substantive new computational contribution beyond prior PPE work. More importantly, the paper assumes a division in which implicit learning is PPE-driven and explicit re-aiming is performance/task-error (PE)-driven, but it does not directly test or rule out the possibility that explicit learning also has a PPE component. For PLOS Computational Biology, where advances in computational concepts are expected, I am not inclined to recommend acceptance in the current form. The dataset itself is strong and may fit better at a venue that emphasizes rigorous empirical reports without a new modeling idea (e.g., Journal of Neurophysiology).

Major comments

1. The paper interprets explicit as PE-driven and implicit as PPE-driven, but there is no head-to-head model test. Please add a formal comparison where the explicit state is driven by (i) PE only, (ii) PPE only, and (iii) PE+PPE, with AIC/BIC (or CV) and parameter plausibility. This will address whether explicit learning carries a PPE component.

2. The modeling framework looks very close to earlier PPE formulations. If there is a conceptual advance, make it explicit (state structure, error-to-state mapping, identifiable regimes, or new theory-level predictions). If not, please position the paper as a replication/validation study and adjust the claims accordingly.

3. Provide simulations (parameter recovery/confusion matrices) showing that your fitting pipeline can recover PPE-only, PE-only, and mixed drivers for explicit and implicit under realistic noise/sample sizes.

4. More direct intervention of PPE is necessary. Consider (or discuss) a manipulation that changes PPE while holding PE constant (e.g., altering visual/proprioceptive uncertainty or localization bias), and the converse. This would strengthen the separation claim.

Minor comments

1. Define and use PPE, PE (task/performance error), SPE, and RPE consistently. A schematic (equations + flow/Vector diagram) contrasting these signals would help.

2. Include full fit summaries (LL, AIC/BIC, ΔAIC/BIC, model weights), parameter CIs/posteriors, and basic diagnostics per experiment.

3. Clarify what is replicated vs extended relative to explicit–implicit separation literature (aim reports, clamped feedback, localization shifts).

Reviewer #3: The authors build new theories, design experiments, and report data on how people adapt to visuomotor rotation. Amongst three models tested, a new “perceptual prediction error” model seems to capture elements of human motor adaptation. This work advances theoretical and experimental understanding of human motor adaptation; the importance of these advances are unclear, due to vagueness and internal contradictions in presentation, calculations, and interpretations.

Strengths:

The manuscript identifies gaps in explanations of human visual motor adaptation, when experiments and analyses separate performance into explicit and implicit components. Previously published work seems to indicate that steady state implicit learning shows either no dependence, or nonlinear dependence, on the strength of perturbation. The authors provide sufficient evidence that two extant theories fail to replicate this finding. A new theory posits that implicit adaptation arises from “perceptual prediction error,” (PPR) the difference between the aiming angle and the angle of the multisensory prediction of hand location. The authors draw from prior work, indicating visual uncertainty increases with perturbation angle. Theorized consequences of that result include Bayesian reweighing of multiple inputs, causing proprioception to increase its relative contribution as visual reliability decreases. From these considerations emerges an overall hypothesis: perceptual prediction errors, sensitive to increased visual noisiness and Bayesian reweighing, could drive implicit learning in visuomotor adaptation.

The manuscript provides solid theoretical bases for this construction, and provides several tests of the PPE model for implicit learning, against models driven by performance error (PE) and (visual) sensory prediction error (SPE). Several analyses indicate that the PPE model better predicts human adaptation, in asymptotic levels of implicit learning, dependence on perturbation strength, and trial-by-trial adaptation. PPE model performance is impressive in its mimicking of nonlinearity with respect to perturbation strength, in both overall implicit learning and trial-by-trial adaptation in the random environment.

The experimental testing of proprioceptive bias, through addition of a bimanual component, provides a particularly strong line of evidence for multisensory consequences of visuomotor rotation, irrespective of particular model choice.

The authors build a comprehensive case for their model, complete with analyzing previously published data; full reporting on their equations, computation methods, and fitted parameters; full online sharing of their code and results; and reference to appropriate literature. The authors provide insights into their reasoning, comparison to prior work, and opportunities for checks and balances through future studies.

Weaknesses:

A major concern arises from the enlarged parameter and composite function space of PPE, compared with PE and SPE models. Analysis of the Bond & Taylor 2015 data generates state space model fits for PE and SPE solely from data, while PPE “as an additional four parameters” (lines 1221-1222). Whereas the rationale for this difference becomes apparent later in the manuscript, and some parameters are fixed from prior work, Figure 2 unfairly compares results arising primarily from data vs. deeper dependence on fits. This nuance is under-explained early in the manuscript and builds confusion for the reader.

A deeper problem arises from the Bayesian calculation of “perceived hand” built from combination of visual cue, proprioceptive cue, and the aim of the reach (line 1126). Aim is not a sensory input; it is a motor goal. While the statistical consequence is not equivalent, in spirit the authors seem to be confusing maximum likelihood and MAP calculations. At the very least, the authors need to justify the direct Bayesian combination of two sensory inputs with a motor goal.

It is confusing how this goal was estimated in trials where subjects did not explicitly report a goal. If the aim angle was estimated in these trials, then the authors have constructed a composite model, which conflates data and simulated inputs to subsequent simulations.

It is confusing how this model is proposed as a core learning rule, when “frequent aiming reports can alter the composition of explicit and implicit learning” (lines 250-251), suggesting deeper cognitive elements of learning than suggested here.

More moderate concerns arise from main text that seems to cover many topics all at once, rather than concentrating on key concepts. Around Figures 2 and 3, individual paragraphs jump across many details of behavior and modeling, generating unclear emphases and conclusions. (A counterexample, featuring very clear organization, is the paragraph starting at line 312 which demonstrates the result and importance of proprioceptive biases.)

Experiments 4 and 5 did add some additional data and comparisons, indicating PPE captured features of learning better than PE and SPE. The cognitive elements, however, provide additional layers of processing, that may partially invalidate motor processing purported to be non-cognitive. The interpretation of a new PE in Experiment 5 is particularly troubling and seems to violate the authors’ own definitions from Figure 1.

Experiment 6 is also validating, but renews the major concern noted above, regarding the estimated nature of aim direction in most of this modeling.

The eye fixation results were somewhat interesting on their own, but do not clearly serve as a “control” for the main paper or as a supplemental set of data. Instead, the fixation data raises concern that hypothesized changes in arm motor processing may instead be a consequence of simultaneous eye and arm control, biasing visual processing.

A minor point: "Re-aiming" may be an appropriate term in the sub-literature, but here, "aiming" would work well. "Re-aiming" infers a mid-reach correction or cognitive override, such as instructed in Experiment 5.

Conclusion:

Overall the authors have proposed interesting ideas, developed relevant experiments, and reported interesting data: the proprioceptive bias results seem to be notable regardless of model choice. There is good evidence that a multisensory signal is more likely to drive difference in implicit learning strength, dependent on perturbation size. The authors should more clearly indicate rationale behind the PPE computations, more clearly focus conclusions and discussions, and more clearly indicate the relevance of late experiments and supplementary material.

**Have the authors made all data and (if applicable) computational code underlying the findings in their manuscript fully available?**

The PLOS Data policy requires authors to make all data and code underlying the findings described in their manuscript fully available without restriction, with rare exception (please refer to the Data Availability Statement in the manuscript PDF file). The data and code should be provided as part of the manuscript or its supporting information, or deposited to a public repository. For example, in addition to summary statistics, the data points behind means, medians and variance measures should be available. If there are restrictions on publicly sharing data or code —e.g. participant privacy or use of data from a third party—those must be specified.requires authors to make all data and code underlying the findings described in their manuscript fully available without restriction, with rare exception (please refer to the Data Availability Statement in the manuscript PDF file). The data and code should be provided as part of the manuscript or its supporting information, or deposited to a public repository. For example, in addition to summary statistics, the data points behind means, medians and variance measures should be available. If there are restrictions on publicly sharing data or code —e.g. participant privacy or use of data from a third party—those must be specified.requires authors to make all data and code underlying the findings described in their manuscript fully available without restriction, with rare exception (please refer to the Data Availability Statement in the manuscript PDF file). The data and code should be provided as part of the manuscript or its supporting information, or deposited to a public repository. For example, in addition to summary statistics, the data points behind means, medians and variance measures should be available. If there are restrictions on publicly sharing data or code —e.g. participant privacy or use of data from a third party—those must be specified.requires authors to make all data and code underlying the findings described in their manuscript fully available without restriction, with rare exception (please refer to the Data Availability Statement in the manuscript PDF file). The data and code should be provided as part of the manuscript or its supporting information, or deposited to a public repository. For example, in addition to summary statistics, the data points behind means, medians and variance measures should be available. If there are restrictions on publicly sharing data or code —e.g. participant privacy or use of data from a third party—those must be specified.

Reviewer #1: Yes

Reviewer #2: None

Reviewer #3: Yes

PLOS authors have the option to publish the peer review history of their article (what does this mean?). If published, this will include your full peer review and any attached files.). If published, this will include your full peer review and any attached files.). If published, this will include your full peer review and any attached files.). If published, this will include your full peer review and any attached files.

...

Reviewer #1: No

Reviewer #2: No

Reviewer #3: No

**Figure resubmission:**

**Reproducibility:**



---

## [Decision Letter · Decision Letter 1]

6 Feb 2026

PCOMPBIOL-D-25-01100R1

Perceptual Prediction Error Supports Implicit Process in Motor Learning

PLOS Computational Biology

Dear Dr. Wei,

Thank you for submitting your manuscript to PLOS Computational Biology. After careful consideration, we feel that it has merit but does not fully meet PLOS Computational Biology's publication criteria as it currently stands. In particular, we feel that the reviewer’s comment about the role of PPE in driving learning should be addressed. Additionally, please consider shortening the manuscript by moving less essential material into the supplementary section. Therefore, we invite you to submit a revised version of the manuscript that addresses the points raised during the review process.

We look forward to receiving your revised manuscript.

Kind regards,

Tianming Yang, Ph.D.

Section Editor

PLOS Computational Biology

Andrea E. Martin

Section Editor

PLOS Computational Biology

**Reviewers' comments:**

Reviewer's Responses to Questions

**Comments to the Authors:**

Reviewer #1: The authors have done an excellent job responding to my concerns. I think this is a rigorous and impactful contribution.

Reviewer #3: The authors provide appropriate responses to reviewer concerns, through additional analyses and figures, and substantive edits to the text. I find these edits answer several of my identified weaknesses. In particular the authors better explain the additional parameters needed to estimate PPE, the efforts to minimize free parameters by relying on previously published values, and the rationale for including aiming direction as an input to Bayesian integration. Overall the manuscript provides evidence that a state space model using PPE to build implicit learning outperforms models using PE and SPE, in capturing the strength of implicit learning across perturbation strengths, training patterns, and additional tasks. Mid-training tests of proprioceptive bias provide additional confirmation that proprioception plays a role in implicit learning during a visuomotor rotation task.

Whereas the additional text, figures, and supplements do address reviewer comments, I worry that the manuscript is now very long. I will leave it to the editors and authors to discern how much content is needed to communicate the central scientific message.

At this point I have one substantive concern. I ask the editors to determine whether the concern is meaningful, and if so, if the authors address the concern. While the PPE model captures changes in implicit learning across perturbation strengths and task demands, I have trouble seeing how the PPE signal constitutes an “error” that drives learning within a training session.

- The dynamics of PPE do not seem appropriate for a learning process. The authors indicate throughout the persistent values of PPE, remaining quite flat throughout training (e.g. Figures 2B and 3B). This time series does not mimic the asymptotic curves traced by implicit learning. The PPE time series that are less flat seem even less related to learning. Initial simulations reveal a time change in early training (Figure 1B). Sizeable stepwise implicit learning could either accompany slopes in PPE (Figure 4B, between trials 7 and 15) or flat PPE (between trials 23 and 39). I acknowledge that the state space models generate quality fits, but it is difficult to process how PPE could be used as a “error signal” to drive these time series of learning.

- The magnitude of PPE does not seem appropriate for the cumulative learning. Throughout the manuscript PPE size is smaller than the accumulated implicit learning. For example, in Figure 3, PPE ranges from -6 to -3 degrees, but modeled implicit learning learning ranges from (roughly) 7 t0 14 degrees. This amplification does not seem plausible, either from casting PPE as an error signal, or from the state space equations with the listed solutions for A and B.

My current conclusion is that proprioception likely plays a role in implicit learning magnitude, and multisensory integration likely serves as a driver of changes in implicit learning across the experimental conditions. I am unclear why or how, however, this construction of PPE can serve as an “error signal” that drives learning, in an abstract sense or as simulated in these state space models.

**Have the authors made all data and (if applicable) computational code underlying the findings in their manuscript fully available?**

The PLOS Data policy requires authors to make all data and code underlying the findings described in their manuscript fully available without restriction, with rare exception (please refer to the Data Availability Statement in the manuscript PDF file). The data and code should be provided as part of the manuscript or its supporting information, or deposited to a public repository. For example, in addition to summary statistics, the data points behind means, medians and variance measures should be available. If there are restrictions on publicly sharing data or code —e.g. participant privacy or use of data from a third party—those must be specified.requires authors to make all data and code underlying the findings described in their manuscript fully available without restriction, with rare exception (please refer to the Data Availability Statement in the manuscript PDF file). The data and code should be provided as part of the manuscript or its supporting information, or deposited to a public repository. For example, in addition to summary statistics, the data points behind means, medians and variance measures should be available. If there are restrictions on publicly sharing data or code —e.g. participant privacy or use of data from a third party—those must be specified.requires authors to make all data and code underlying the findings described in their manuscript fully available without restriction, with rare exception (please refer to the Data Availability Statement in the manuscript PDF file). The data and code should be provided as part of the manuscript or its supporting information, or deposited to a public repository. For example, in addition to summary statistics, the data points behind means, medians and variance measures should be available. If there are restrictions on publicly sharing data or code —e.g. participant privacy or use of data from a third party—those must be specified.requires authors to make all data and code underlying the findings described in their manuscript fully available without restriction, with rare exception (please refer to the Data Availability Statement in the manuscript PDF file). The data and code should be provided as part of the manuscript or its supporting information, or deposited to a public repository. For example, in addition to summary statistics, the data points behind means, medians and variance measures should be available. If there are restrictions on publicly sharing data or code —e.g. participant privacy or use of data from a third party—those must be specified.

Reviewer #1: Yes

Reviewer #3: Yes

PLOS authors have the option to publish the peer review history of their article (what does this mean?). If published, this will include your full peer review and any attached files.). If published, this will include your full peer review and any attached files.). If published, this will include your full peer review and any attached files.). If published, this will include your full peer review and any attached files.

...

Reviewer #1: No

Reviewer #3: No

**Figure resubmission:**
---

## [Editor Report · Decision Letter 2]

2 Apr 2026

Dear Dr Wei,

We are pleased to inform you that your manuscript 'Perceptual Prediction Error Supports Implicit Process in Motor Learning' has been provisionally accepted for publication in PLOS Computational Biology.

Best regards,

CompBiol Staff

Journal Office CompBiol

PLOS Computational Biology

Andrea E. Martin

Section Editor

PLOS Computational Biology

---

## [Editor Report · Acceptance letter]

PCOMPBIOL-D-25-01100R2

Perceptual Prediction Error Supports Implicit Process in Motor Learning

Dear Dr Wei,

I am pleased to inform you that your manuscript has been formally accepted for publication in PLOS Computational Biology. Your manuscript is now with our production department and you will be notified of the publication date in due course.

With kind regards,

Anita Estes
